# BRAIN PATHOGRAPH LEARNING

## ABSTRACT

Brain graph learning has demonstrated significant achievements in the fields of neuroscience and artificial intelligence. However, existing methods struggle to selectively learn disease-related knowledge, leading to heavy parameters and computational costs. This challenge diminishes their efficiency, as well as limits their practicality for real-world clinical applications. To this end, we propose a lightweight Brain PathoGraph Learning (BrainPoG) model that enables efficient brain graph learning by pathological pattern filtering and pathological feature refinement. Specifically, BrainPoG first contains a filter to extract the pathological pattern formulated by highly disease-relevant subgraphs, achieving graph pruning and lesion localization. A PathoGraph is therefore constructed by dropping less disease-relevant subgraphs from the whole brain graph. Afterwards, a pathological feature refinement module is designed to reduce disease-irrelevant noise features and enhance pathological features of each node in the PathoGraph. BrainPoG can exclusively learn informative disease-related knowledge while avoiding less relevant information, achieving efficient brain graph learning. Extensive experiments on four benchmark datasets demonstrate that BrainPoG exhibits superiority in both model performance and computational efficiency across various brain disease detection tasks[1].

## 1 INTRODUCTION

Brain graph learning is pivotal at the intersection of neuroscience and artificial intelligence, aiming to model the complex structural and functional connectivity of the nervous system and to support various clinical tasks (Seguin et al., 2023; Cho et al., 2024; Yang et al., 2025). Brain functional graphs (BFGs), serving as one of the most fundamental representations of brain graphs, provide critical insights into the complex functional connectivity within the brain (Kan et al., 2022; Yu et al., 2024; Peng et al., 2025). Typically derived from functional magnetic resonance imaging (fMRI), BFGs capture functional correlations between the blood-oxygen-level-dependent signals of paired brain regions of interest (ROIs), offering information closely associated with psychiatric and neurological conditions. Therefore, learning from BFGs can support clinical tasks such as the detection of disease-related abnormalities, which is particularly valuable for understanding pathological conditions in the brain (Luo et al., 2024; Xu et al., 2025).

Notably, lightweight design in brain graph learning models is essential, especially for privacy-sensitive and low-resource clinical deployment. In practice, such models often need to run on on-premise clinical workstations or local hospital servers, which typically have limited GPUs and restricted computational capacity due to strict medical data privacy regulations. Lightweight models enable on-site processing without transmitting sensitive brain data outside the hospital network, thereby ensuring both data privacy and computational efficiency. However, existing brain graph learning techniques still suffer from non-trivial computational overhead, and this efficiency issue remains a key challenge for practical clinical applicability. Existing methods often indiscriminately learn from the whole brain graph, which contains substantial disease-irrelevant noise, *lacking the ability to pinpoint critical pathological patterns and pathological features*. Particularly, in the absence of precise biomarkers, they often provide insufficient insights into disease-specific alterations in the brain and fail to achieve lesion localization (Mahesh & Tasneem, 2014; Le et al., 2025). This limitation not only undermines model performance in downstream tasks but also necessitates larger model architectures to offset the lack of pathological focus. Therefore, learning from complex brain

---

[1]The code is available at: https://anonymous.4open.science/r/BrainPoG-6F5E

graph data often entails *high parameter demands and substantial computational overhead* (Li et al., 2023; Zong et al., 2024). For example, ALTER (Yu et al., 2024) requires 4.6M parameters and 2278MB of memory when trained on the Autism Brain Imaging Data Exchange (ABIDE)[2] dataset, while BRAINNETTF (Kan et al., 2022) uses over 4.0M parameters and 2150MB of memory on ABIDE. Such requirements hinder the applicability of these models and, in particular, limit their practical deployment in time-sensitive or resource-constrained settings.

To tackle this challenge, we propose Brain PathoGraph Learning (**BrainPoG**), a lightweight model that is designed to avoid learning disease-irrelevant knowledge and instead focus solely on informative pathological patterns and pathological features, thereby enabling efficient brain graph learning. BrainPoG first contains a *pathological pattern filter* to select highly disease-relevant subgraphs from the whole brain graph. Specifically, we introduce a subgraph classifier, which evaluates each subgraph with a patho-score, a quantitative measure that evaluates the disease-relevance of a subgraph within the whole brain graph. A high patho-score indicates that the subgraph is highly relevant to the disease, whereas a low patho-score suggests limited relevance. We then introduce *PathoGraph*, which is constructed by dropping less disease-relevant subgraphs (i.e., those with low patho-scores) from the whole brain graph. Therefore, the pathological pattern filter is designed to filter out less disease-relevant subgraphs and form the PathoGraph, achieving graph pruning and lesion localization. Afterwards, we design a *pathological feature refinement module* to drop disease-irrelevant noise features and enhance pathological features of each node in the PathoGraph. Disease-irrelevant noise features refer to node features that are not disease-specific and contribute little to disease classification. This module enables BrainPoG to further focus on disease-specific feature learning, yield more discriminative brain graph representations. Finally, a simple graph convolutional network (GCN) (Kipf & Welling, 2017) trained with cross-entropy loss is applied to the PathoGraphs with enhanced node features.

**Contributions.** Building upon the unique design, BrainPoG establishes a lightweight yet effective architecture that not only enables disease-specific feature learning but also substantially reduces model size and computational overhead. This enhances the practicality of BrainPoG for real-world clinical applications. Our main contributions are as follows: **(1)** We propose a novel Brain Patho-Graph Learning (BrainPoG) model, which is designed to enable efficient brain graph learning. **(2)** We introduce the concept of a PathoGraph and design a pathological pattern filter to achieve graph pruning and facilitating lesion localization. **(3)** We present a pathological feature refinement method to enhance pathological features, enabling more discriminative brain graph representations. **(4)** We conduct extensive experiments on four benchmark datasets, demonstrating that BrainPoG outperforms state-of-the-art methods in model performance and computational efficiency.

## 2 RELATED WORK

**Brain Graph Learning.** Brain functional graphs constructed from fMRI reflect the functional connectivity of the human neural system (Bessadok et al., 2022; Qiu et al., 2024; Peng et al., 2025; Tang et al., 2022). In brain functional graphs, nodes represent brain regions of interest (ROIs) and edges are functional correlations between ROIs (Said et al., 2024; Cui et al., 2022). With the growing success of graph learning in modeling complex relationships between entities, brain graph learning has emerged as one of the most promising method for various brain graph analytics tasks, such as brain disease detection (Kan et al., 2022; Ding et al., 2023; Yu et al., 2024). A variety of advanced brain graph learning models have been proposed for different objectives (Li et al., 2023; Gu et al., 2025; Zhang et al., 2024; Tang et al., 2022). For example, BrainGNN (Li et al., 2021) captures the functional information of brain graphs constructed from fMRI by ROI-aware graph convolutional layers. STAGIN (Kim et al., 2021) uses a spatiotemporal attention mechanism in graph neural networks to learn the dynamic brain graph representation, capturing the temporal changes in brain graphs. BRAINNETTF (Kan et al., 2022) presents a Transformer-based model for brain graph analysis, achieving advanced brain disease detection tasks. MSE-GCN (Lei et al., 2023) encodes multimodal brain graphs applies by multiple parallel graph convolutional network layers for early Alzheimer's disease (AD) detection. GroupBNA (Peng et al., 2024) presents a group-adaptive brain network augmentation strategy to construct group-specific brain graphs for accurate brain disease detection. ALTER (Yu et al., 2024) utilizes biased random walks to capture long-range dependencies

---

[2] https://fcon_1000.projects.nitrc.org/indi/abide/

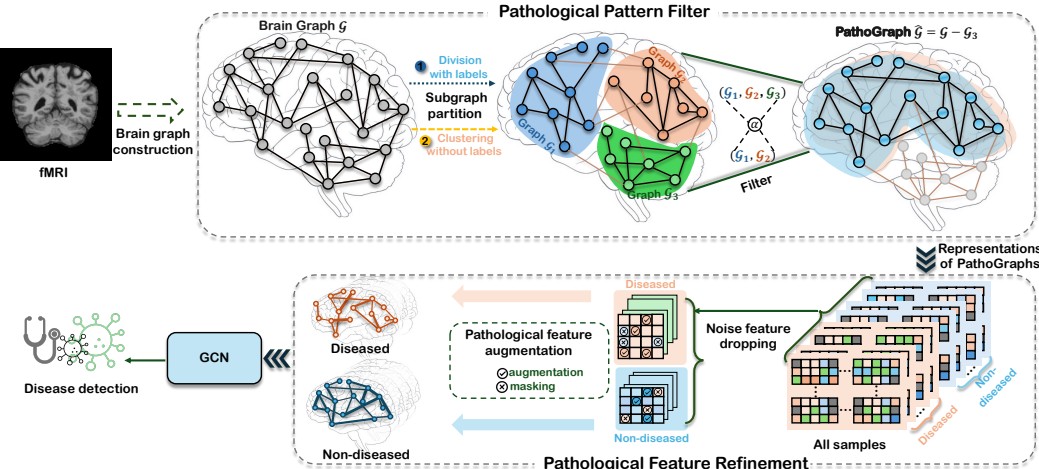

Figure 1: The overall framework of BrainPoG.

between brain ROIs. BioBGT (Peng et al., 2025) encodes the small-world architecture of brain graphs, therefore enhancing the biological plausibility of the learned brain graph representations.

Although many current brain graph learning methods show promising performance, they often lack the capability to selectively learn disease-related knowledge, compromising efficiency in downstream tasks (Le et al., 2025). Therefore, they suffer from high parameter requirements and computational overhead, which impedes the practical deployment of these models in time-sensitive or resource-constrained settings. Some works have explored strategies to improve model efficiency. For example, IGS (Li et al., 2023) improves model efficiency by iteratively eliminating noisy edges to address the issue of dense brain graphs. However, the challenge of simultaneously maintaining model performance while reducing model size and computational cost remains largely unresolved in brain graph learning. To fill this gap, we propose BrainPoG, a lightweight brain graph learning framework that explicitly learns from pathological patterns and pathological features to achieve efficient brain graph learning.

## 3 METHOD

In this section, we introduce our BrainPoG in detail. The overall framework of BrainPoG is illustrated in Figure 1. BrainPoG contains two main modules: (1) pathological pattern filter and (2) pathological feature refinement. The pathological pattern filter is first designed to filter out less disease-relevant subgraphs and construct the PathoGraph. Then, we propose a pathological feature refinement to drop noise features and enhance pathological features of each node in the PathoGraph.

### 3.1 PATHOLOGICAL PATTERN FILTER

We first construct brain graphs based on fMRI data. Particularly, the functional correlations (e.g., edges between nodes) are the computed Pearson correlation coefficient (PCC) (Cohen et al., 2009) between ROIs. For a given brain graph, it is denoted as $\mathcal{G} = (\mathcal{V}, \mathbf{A})$, where $\mathcal{V}$ stands for the node (i.e., ROI) set and $\mathbf{A} \in \mathbb{R}^{N \times N}$ is the adjacency matrix. $N$ represents the number of nodes. The node representation $\mathbf{X} \in \mathbb{R}^{N \times N}$ is directly derived from the adjacency matrix, where each node is represented by its corresponding adjacency vector (Kan et al., 2022). We first partition the whole brain graph into several subgraphs, each of which represents a functional module of the brain, denoted as $\{\mathcal{G}_1, \mathcal{G}_2, ..., \mathcal{G}_m\}$, where $\mathcal{G}_i$ indicates the $i$-th subgraph and $m$ is the number of subgraphs. Particularly, based on different brain parcellation schemes, we adopt different strategies for subgraph partition, as some parcellations provide ROI-level labels while others do not. For graphs with labeled nodes, we partition subgraphs using well-established neuroscience priors. For example, the brain graph constructed upon the AAL atlas (Tzourio-Mazoyer et al., 2002) contains 90 labeled ROIs (i.e., nodes) and can be divided into 9 subgraphs, each representing a distinct functional module. Table 4 in Appendix B shows the detailed subgraph partition for AAL atlas-based brain graphs. In contrast, for graphs without labeled nodes, such as those constructed based on the Craddock 200

atlas (Craddock et al., 2012), we obtain subgraphs by applying a community detection method, specifically spectral clustering (Fortunato, 2010), to identify functionally coherent node groups.

Then, we design a filter to extract the pathological pattern formulated by highly disease-relevant subgraphs. Specifically, we implement pathological pattern filtering using a support vector machine (SVM)-based subgraph classifier. We first train an SVM with a radial basis function (RBF) kernel on the brain graph dataset, where each sample corresponds to a complete brain graph $\mathcal{G}$. The SVM outputs a classification score (e.g., the accuracy of graph classification task), denoted as $\alpha$. Afterwards, for each subgraph $\mathcal{G}_i$, we construct a subgraph-only dataset by retaining only $\mathcal{G}_i$ from each sample. This subgraph-specific dataset is then fed into the SVM to obtain the classification score, which is denoted as its patho-score $\beta_i$. To ensure statistical rigor, both the classification score $\alpha$ and each patho-score $\beta_i$ are computed as the mean over five independent SVM runs with different random seeds. This averaging strategy reduces variance and ensure that the resulting patho-scores are statistically more reliable. Consequently, we can get the set of patho-scores $\beta = \{\beta_1, \beta_2, ..., \beta_m\}$ for all subgraphs, which quantitatively reflect their disease relevance. If $\beta_i \geq \alpha$, subgraph $\mathcal{G}_i$ is considered essential for the disease; otherwise, it is regarded as less relevant. Based on this criterion, we construct the PathoGraph $\hat{\mathcal{G}}$ by dropping subgraphs with lower path-scores (e.g., $\beta_i < \alpha$) and retaining those yielding higher classification performance in the complete brain graph. Here, we give the definition of the PathoGraph.

**Definition 1** (PathoGraph). *The PathoGraph $\hat{\mathcal{G}}$ is constructed by dropping less disease-relevant subgraphs from the complete brain graph $\mathcal{G}$, denoted as $\hat{\mathcal{G}} = \mathcal{G} - \{\mathcal{G}\}_{i=1}^{\hat{m}}$. PathoGraph represents the pathological pattern of the brain disease extracted from the complete brain graph.*

Here, $\hat{m}$ indicates the number of dropped subgraphs. By filtering out subgraphs that are less relevant to the disease and obtaining the informative PathoGraph, we not only achieve lesion localization but also effectively achieve graph pruning. Algorithm 1 (in Appendix C) illustrates the procedure of our pathological pattern filter.

## 3.2 PATHOLOGICAL FEATURE REFINEMENT

**Noise Feature Dropping.** The node representation of PathoGraph $\hat{\mathcal{G}}$ is denoted as $\hat{\mathbf{X}} \in \mathbb{R}^{\hat{N} \times \hat{N}}$, where $\hat{N} \ll N$ is the number of retained nodes after filtering. For all $D$ samples in the PathoGraph dataset $\hat{\mathcal{B}}$, we use singular value decomposition (SVD)-based scoring function to identify communal features across all samples. Communal features, considered as a type of disease-irrelevant noise features, are features that commonly exist across most subjects and lack discriminative power for disease differentiation. Therefore, communal features are dropped to reduce redundancy.

Given the node representation set of $\hat{\mathcal{B}}$, $\{\hat{\mathbf{X}}^{(1)}, \hat{\mathbf{X}}^{(2)}, ..., \hat{\mathbf{X}}^{(D)}\}$, we first rearrange the features of each node across all $D$ subjects to construct cross-subject feature representations. Formally, for node $v_i$, its cross-subject feature representation is $\mathbf{C}_{v_i} \in \mathbb{R}^{\hat{N} \times D}$. Then, we perform SVD on the cross-subject feature representations to evaluate the significance of each feature dimension of each node. The first column of the left singular matrix obtained from SVD is used as the scoring criterion for identifying communal features across all subjects. Therefore, a communal feature score matrix $\mathbf{S} \in \mathbb{R}^{\hat{N} \times \hat{N}}$ for all nodes can be obtained:

$$
\begin{aligned}
\mathbf{C}_{v_i} &= \mathbf{L}_{v_i} \boldsymbol{\Sigma}_{v_i} \mathbf{R}_{v_i}^{\top}, \quad v_i \in \hat{N}, \\
\mathbf{l}_{v_i} &= \mathbf{L}_{v_i}[:, 1] = \left[\mu_{v_i}^1, \mu_{v_i}^2, \ldots, \mu_{v_i}^{\hat{N}}\right]^{\top}, \\
\mathbf{S} &= [\mathbf{l}_{v_1}, \mathbf{l}_{v_2}, \ldots, \mathbf{l}_{v_{\hat{N}}}]^{\top}.
\end{aligned}
\tag{1}
$$

Here, $\mathbf{L}_{v_i}$ and $\mathbf{R}_{v_i}$ indicate the left and right singular matrices of node $v_i$, respectively, and $\boldsymbol{\Sigma}_{v_i}$ is the diagonal matrix. $\mathbf{l}_{v_i}$ is the first left singular vector and stands for the feature score vector of node $v_i$, where $\mu_{v_i}^j$ is the score of the $j$-th feature. To identify communal features across all subjects, we rank the features of each node based on their feature scores.

**Proposition 1** (Communal features). *Based on feature scores, the top ranked features in the cross-subject feature representation $\mathbf{C}_{v_i}$ are considered as communal features of node $v_i$ across all subjects.*

The proof of Proposition 1 is given in Appendix D. Specifically, the top-$k$ ranked features are considered as communal features across all subjects and are dropped to achieve feature refinement and eliminate redundancy. In addition, we also drop the bottom-$k$ features due to their limited contributions to informative node representations. Afterwards, for the PathoGraph $\hat{\mathcal{G}}$, we can get its distilled node representation, denoted as $\widetilde{\mathbf{X}} \in \mathbb{R}^{\hat{N} \times \hat{N}'}$, where $\hat{N}' \ll \hat{N}$ is determined by the value of $k$.

**Pathological Feature Augmentation.** After dropping noise features, we further enhance the pathological features (e.g., group-specific features within each subject group) to strengthen their discriminative representations. Therefore, pathological feature augmentation refers to amplifying features that show significant inter-group variations. For example, in the Attention Deficit Hyperactivity Disorder (ADHD-200) dataset, samples can be divided into the normal control group and the ADHD patient group. We separately enhance pathological features in the ADHD patient group and features in the NC group, highlighting the disease-specific features.

For the dataset $\hat{\mathcal{B}}$, subjects are divided into $Y$ groups based on the given labels. We apply the same SVD-based scoring function with Equation (1) to the node representations of each group independently, obtaining the group-specific feature score matrices $\{\mathbf{F}_1, \mathbf{F}_2, ..., \mathbf{F}_Y\}$, where $\mathbf{F}_y \in \mathbb{R}^{\hat{N} \times \hat{N}'}$ is the group-specific feature score matrix for group $y$. Features with high group-specific scores are considered as important pathological features. Then, we introduce a feature augmentation strategy based on group-specific feature score matrices. Compared to existing feature augmentation methods (Zhu et al., 2021; Zhang et al., 2023b), our approach incorporates a feature masking operation guided by the group-specific feature scores. Specifically, features with higher group-specific scores are retained, while less informative features are masked out to enhance group-discriminative representation. The feature weights are defined as:

$$\mathbf{W} = \frac{1}{\hat{N}} \sum_{i=1}^{\hat{N}} |\widetilde{\mathbf{x}}_i| \odot \mathbf{F}_{y,i}, \tag{2}$$

where $\widetilde{\mathbf{x}}_i \in \mathbb{R}^{\hat{N}'}$ is the feature vector of node $v_i$, and $\mathbf{F}_{y,i} \in \mathbb{R}^{\hat{N}'}$ represents the group-specific score vector of node $v_i$ in group $y$. $\odot$ denotes element-wise multiplication. $\mathbf{W} = [w_1, w_2, ..., w_{\hat{N}'}]$ contains the weights for each feature dimension. Then, for the $j$-th feature, we calculate the masking probability according to its importance (weight) obtained from Equation (2):

$$p_j = \min\left(\rho \cdot \frac{\log w_{max} - \log w_j}{\log w_{max} - \log \lambda_w}, p_t\right). \tag{3}$$

Here, $\rho$ is a hyperparameter adjusting the magnitude of feature enhancement, $w_{max}$ and $\lambda_w$ are the maximum and average weight values of all feature dimensions, and $p_t < 1$ is a truncation threshold that prevents overly aggressive deletion. We then sample a binary mask vector $\tilde{\mathbf{b}} \in \{0, 1\}^{\hat{N}'}$, where each dimension element is sampled from a Bernoulli distribution:

$$\tilde{b}_j \sim \text{Bernoulli}(1 - p_j), \quad \forall j = 1, 2, \ldots, \hat{N}'. \tag{4}$$

By sampling the binary mask from the Bernoulli distribution, we can more robustly identify informative features (Lee et al., 2022). Finally, the enhanced node representation is computed as:

$$\widetilde{\mathbf{X}}' = \left[\widetilde{\mathbf{x}}_1 \odot \tilde{\mathbf{b}}; \widetilde{\mathbf{x}}_2 \odot \tilde{\mathbf{b}}; ...; \widetilde{\mathbf{x}}_N \odot \tilde{\mathbf{b}}\right]^\top. \tag{5}$$

Algorithm 2 (in Appendix C) shows the procedure of the pathological feature refinement. After obtaining the enhanced node representations of all PathoGraphs, we fed them into a simple GCN model. Specifically, we employ the cross-entropy loss function to optimize the GCN model.

## 4 EXPERIMENTS

### 4.1 EXPERIMENTAL SETUP

**Datasets.** We conduct experiments on fMRI data collected from four benchmark datasets. (1) Alzheimer's Disease Neuroimaging Initiative (ADNI) [3] dataset contains 407 subjects, including 190

---
[3]https://adni.loni.usc.edu/

normal controls (NCs), 170 mild cognitive impairment (MCI) patients, and 47 Alzheimer's disease (AD) patients. (2) Parkinson's Progression Markers Initiative (PPMI) dataset (Marek et al., 2018) includes fMRI data of 49 Parkinson's disease (PD) patients, 69 individuals at risk for PD (Prodromal), and 40 NCs. (3) Attention Deficit Hyperactivity Disorder (ADHD-200) [4] dataset comprises a total of 459 subjects in which 230 subjects are typically developing individuals and 229 subjects are ADHD patients. (4) Autism Brain Imaging Data Exchange (ABIDE) [5] dataset contains 1, 009 subjects including 516 Autism spectrum disorder patients and 493 NCs. The definition of ROI in ADNI and PPMI datasets is based on AAL atlas (Tzourio-Mazoyer et al., 2002). Therefore, graphs in these two datasets have ROI-level labels, and their subgraphs can be obtained by division with labels (see Table 4). The ROIs of graphs in ADHD-200 and ABIDE datasets are defined by Craddock 200 atlas (Craddock et al., 2012), which does not provide ROI-level labels. Thereby, we apply spectral clustering to generate subgraphs for these two datasets. The number of ROIs in ADNI and PPMI is 90. For ABIDE and ADHD-200, the numbers of ROIs are 200 and 190, respectively.

**Evaluation Metrics.** We evaluate our model on the disease detection task, which is regarded as graph classification problem. Notably, for ABIDE and ADHD-200 datasets, the classification task is a binary classification problem, detecting whether the subject is a patient or a normal control. For ADNI and PPMI datasets, which have three groups, the disease detection is a multiple classification problem. To evaluate the model performance, we choose three metrics, including accuracy (ACC), F1 score, area under the receiver operating characteristic curve (AUC). For the multiclass classification task on ADNI and PPMI datasets, we use macro averaging for the F1 score. In addition, to evaluate the model efficiency, we calculate the parameter number, running time, and memory usage.

**Baseline Methods.** We compare our model with state-of-the-art brain graph learning models, including BrainGB (Cui et al., 2022), BRAINNETTF (Kan et al., 2022), A-GCL (Zhang et al., 2023a), MCST-GCN (Zhu et al., 2024), ALTER (Yu et al., 2024), BioBGT (Peng et al., 2025), Brain-OOD (Xu et al., 2025), and HGST (Han et al., 2026). In addition, we compare our method with two representative graph learning baselines, GCN (Kipf & Welling, 2017) and graph attention network (GAT) (Veličković et al., 2018), as well as two typical machine learning approaches, SVM and random forest (RF).

**Implementation Details.** Our model is implemented using PyTorch Geometric v2.6.1, PyTorch v2.7.0, and NetworkX v3.4.2. Model training is performed on an NVIDIA A10G GPU with 64GB of memory. We train our model with the Adam optimizer and use cross-entropy loss for classification. For evaluation, we perform stratified five-fold cross-validation, in each fold, the dataset is randomly partitioned into 70% training, 10% validation, and 20% test sets. Full implementation is given in Appendix E.

## 4.2 RESULTS

**Model Performance Comparison.** Table 1 compares the results of BrainPoG with baselines. As shown by the experimental results, BrainPoG consistently outperforms other baselines across all four datasets. Notably, BrainPoG achieves the highest accuracy of 83.31%, 82.90%, 93.16%, and 93.03% on four datasets, respectively, representing improvements of 7.88%, 2.03%, 8.83%, and 6.91% over the second-best methods. Given the binary-classification nature of ABIDE and ADHD-200, higher accuracy is expected on those two datasets. Beyond accuracy, BrainPoG also demonstrates the highest AUC and F1 across four datasets, achieving AUCs of 92.23%, 92.00%, 98.92% and 97.77%, and F1 scores of 69.86%, 81.22%, 93.13%, and 93.03%, respectively, indicating large gains in classwise discrimination. The experimental results demonstrate that our model excels in various brain disease detection tasks.

**Model Efficiency Analysis.** In addition to evaluating model performance, it is crucial to assess the computational efficiency of the proposed method. To this end, we report the number of parameters, running time, and memory usage of BrainPoG during training across all selected datasets. As summarized in Table 2, BrainPoG requires fewer parameters, shorter running time, and lower

---

[4] https://fcon_1000.projects.nitrc.org/indi/adhd200/
[5] https://fcon_1000.projects.nitrc.org/indi/abide/

Table 1: Experimental results on four datasets (%). The best results are marked in **bold**, and the suboptimal results are marked underlined.

| Method | ADNI | | | PPMI | | | ABIDE | | | ADHD-200 | | |
|---|---|---|---|---|---|---|---|---|---|---|---|---|
| | ACC | AUC | F1 | ACC | AUC | F1 | ACC | AUC | F1 | ACC | AUC | F1 |
| SVM | 51.37±2.81 | 61.02±3.21 | 35.28±2.03 | 41.00±4.06 | 55.04±3.16 | 19.47±1.30 | 57.00±7.65 | 71.73±2.58 | 50.43±11.00 | 58.20±7.66 | 70.10±2.55 | 51.21±7.33 |
| RF | 55.69±4.45 | 62.76±5.54 | 39.28±3.13 | 37.50±6.89 | 49.42±5.80 | 27.84±8.72 | 53.04±3.74 | 56.36±3.97 | 52.93±3.80 | 65.22±2.64 | 75.49±1.64 | 65.16±2.65 |
| GCN | 51.43±1.90 | 55.31±3.39 | 35.49±1.35 | 45.88±11.41 | 58.24±6.14 | 32.76±4.66 | 55.29±4.87 | 68.23±2.57 | 53.82±5.00 | 62.13±7.17 | 74.09±5.73 | 61.10±7.37 |
| GAT | 51.95±5.43 | 61.18±4.44 | 39.38±6.47 | 55.00±1.53 | 63.72±3.46 | 44.99±5.79 | 58.71±1.99 | 60.98±3.40 | 57.05±2.56 | 59.35±4.22 | 63.37±4.69 | 58.66±4.01 |
| BrainGB | 54.88±2.44 | 65.16±3.66 | 43.92±3.47 | 31.87±11.76 | 50.00±4.10 | 37.81±11.66 | 62.87±1.00 | 70.42±8.40 | 62.94±2.20 | 65.22±1.23 | 71.34±5.30 | 65.15±7.20 |
| BRAINNETTF | 70.43±3.21 | 77.65±2.42 | 57.54±2.87 | 80.87±4.01 | 87.43±3.01 | 77.30±3.45 | 77.33±1.04 | 84.27±2.18 | 76.86±4.23 | 75.12±1.67 | 82.56±2.34 | 75.03±1.09 |
| A-GCL | 56.43±2.12 | 60.12±2.23 | 47.76±3.56 | 57.54±2.87 | 61.23±2.45 | 56.12±3.76 | 62.65±2.88 | 68.42±2.85 | 62.14±2.96 | 65.12±4.45 | 70.22±4.28 | 64.79±4.51 |
| MCST-GCN | 75.43±2.54 | 80.65±2.10 | 61.54±2.81 | 79.87±4.01 | 85.43±2.94 | 77.34±3.40 | 84.33±1.04 | 87.10±2.18 | 84.26±4.23 | 86.12±1.67 | 91.56±2.34 | 85.73±1.09 |
| ALTER | 66.38±5.51 | 77.65±4.58 | 68.87±4.71 | 64.29±5.05 | 83.38±2.66 | 71.36±8.62 | 71.20±1.30 | 76.60±1.13 | 72.43±1.55 | 74.05±8.02 | 82.62±5.63 | 74.55±6.84 |
| BioBGT | 55.11±2.34 | 62.48±5.02 | 53.89±2.64 | 54.57±4.69 | 65.20±6.03 | 67.20±1.76 | 74.41±2.32 | 73.33±2.34 | 68.43±2.11 | 71.90±1.28 | 70.75±1.76 | 75.31±1.06 |
| BrainOOD | 68.10±3.97 | 69.84±3.89 | 65.23±5.35 | 66.10±5.42 | 64.93±6.02 | 62.12±3.33 | 66.24±5.12 | 64.21±3.29 | 63.89±4.20 | 67.43±4.47 | 66.59±3.75 | 65.39±4.11 |
| HGST | 57.33±2.77 | 61.02±3.15 | 54.76±2.58 | 54.06±2.20 | 64.33±3.10 | 53.50±2.13 | 68.39±2.40 | 68.15±2.58 | 68.20±2.52 | 66.89±4.05 | 66.94±3.96 | 66.30±4.14 |
| **BrainPoG** | **83.31±4.90** | **92.23±2.83** | **69.86±8.21** | **82.90±1.62** | **92.00±1.50** | **81.22±1.66** | **93.16±2.46** | **98.92±0.67** | **93.13±2.49** | **93.03±3.27** | **97.77±2.19** | **93.03±3.27** |

computational overhead than state-of-the-art brain graph learning models. By contrast, simple models such as SVM, RF, and GCN indeed use a few of parameters and computational costs but suffer from significantly worse model performance, whereas methods that achieve accuracy comparable to BrainPoG demand more parameters and computational overhead. These results demonstrate that BrainPoG yields substantial improvements in computational efficiency while improving model performance. In addition, to further verify the efficiency of BrainPoG, we compare its time and space complexity with other state-of-the-art brain graph learning models (see Table 7 in Appendix F.3).

Table 2: Model efficiency comparison.

| Method | ADNI | | | PPMI | | | ABIDE | | | ADHD-200 | | |
|---|---|---|---|---|---|---|---|---|---|---|---|---|
| | Parameter No. | Running time | Memory | Parameter No. | Running time | Memory | Parameter No. | Running time | Memory | Parameter No. | Running time | Memory |
| SVM | 579 | 0.0332s/run | 0.37MB | 239 | 0.0079s/run | 0.15MB | 238 | 0.0078s/run | 0.14MB | 238 | 0.0078s/run | 0.14MB |
| RF | 11K | 0.4463s/run | 0.24MB | 7K | 0.3026s/run | 0.13MB | 20K | 1.0445s/run | 0.84MB | 9K | 0.5158s/run | 0.40MB |
| GCN | 8K | 8.6810s/epoch | 74MB | 8K | 5.9670s/epoch | 26MB | 15K | 36.1985s/epoch | 698MB | 14K | 16.8647s/epoch | 424MB |
| GAT | 52K | 0.8165s/epoch | 1123MB | 52K | 0.4017s/epoch | 1182MB | 108K | 1.9470s/epoch | 1278MB | 103K | 0.8945s/epoch | 1186MB |
| BrainGB | 628K | 2.7192s/epoch | 1219MB | 628K | 1.1525s/epoch | 1229MB | 981K | 31.2101s/epoch | 1807MB | 898K | 11.6271s/epoch | 1517MB |
| BRAINNETTF | 1.8M | 10.2300s/epoch | 1752MB | 1.8M | 6.3400s/epoch | 1700MB | 4.0M | 42.5600s/epoch | 2150MB | 3.8M | 18.7623s/epoch | 1970MB |
| A-GCL | 246K | 3.5060s/epoch | 450MB | 246K | 1.5543s/epoch | 380MB | 468K | 8.5600s/epoch | 750MB | 448K | 5.2800s/epoch | 620MB |
| MCST-GCN | 1.3M | 5.8392s/epoch | 1550MB | 1.3M | 5.0323s/epoch | 1522MB | 3.8M | 39.8904s/epoch | 2088MB | 3.7M | 15.9895s/epoch | 1788MB |
| ALTER | 1.6M | 9.2993s/epoch | 1580MB | 1.4M | 4.8729s/epoch | 1510MB | 4.6M | 44.5332s/epoch | 2278MB | 4.3M | 23.0471s/epoch | 2109MB |
| BioBGT | 455K | 8.5322s/epoch | 1023MB | 338K | 3.2800s/epoch | 422MB | 465K | 15.300s/epoch | 917MB | 454K | 5.2150s/epoch | 590MB |
| BrainOOD | 469K | 6.3000s/epoch | 846MB | 391K | 2.5810s/epoch | 380MB | 528K | 11.4728s/epoch | 872MB | 496K | 6.4000s/epoch | 574MB |
| HGST | 10M | 112.4500s/epoch | 2589MB | 8.4M | 89.3210s/epoch | 2231MB | 15M | 454.5000s/epoch | 7629MB | 24M | 380.4100s/epoch | 3846MB |
| **BrainPoG** | 227K | 0.0046s/epoch | 23MB | 11K | 0.0029s/epoch | 17MB | 415K | 0.0030s/epoch | 99MB | 140K | 0.0030s/epoch | 23MB |

**PathoGraph Identification and Lesion Localization.** To evaluate the effectiveness of our pathological pattern filter, Figure 2 presents the patho-scores obtained to identify PathoGraph for four different diseases. As illustrated, only a few of subgraphs yield patho-scores higher than the classification score $\alpha$. For Alzheimer's disease, the PathoGraph is formed by remaining only the 'Limbic' and 'Subcortical' subgraphs (Figure 2a), in line with prior neuroscience findings (Nelson et al., 2019; Yang et al., 2021). For Parkinson's disease, the 'Cognitive Control', 'DMN' (default mode network), and 'Subcortical' subgraphs are identified as highly disease-relevant and remained to form the PathoGraph (Figure 2b), a finding highly consistent with existing neuroscience research (Lu et al., 2024). Given the absence of predefined region-of-interest (ROI) labels in the ABIDE and ADHD-200 brain graphs, we derived the PathoGraphs by identifying unlabeled subgraphs that yielded high patho-scores for the target disorders, namely Autism Spectrum Disorder and ADHD (Figures 2c and 2d). Meanwhile, we visualize the lesions identified by the filter for four diseases (Figure 3). The results indicate that the filter effectively eliminates less relevant regions and retains highly disease-relevant regions, enabling lesion localization. In addition, to demonstrate BrainPoG's ability to reveal disease-specific alterations, we visualize and compare group-wise heatmaps of PathoGraph representations. The results are shown in Figure 10 and Figure 11 in Appendix F.4.

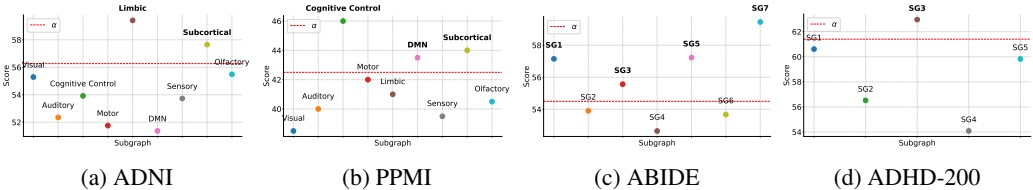

| (a) ADNI | (b) PPMI | (c) ABIDE | (d) ADHD-200 |

Figure 2: PathoGraph identification for four different diseases. The PathoGraph is constructed by remaining subgraphs shown in **bold** and dropping those having lower patho-scores than $\alpha$. 'SG' denotes the subgraph without labels.

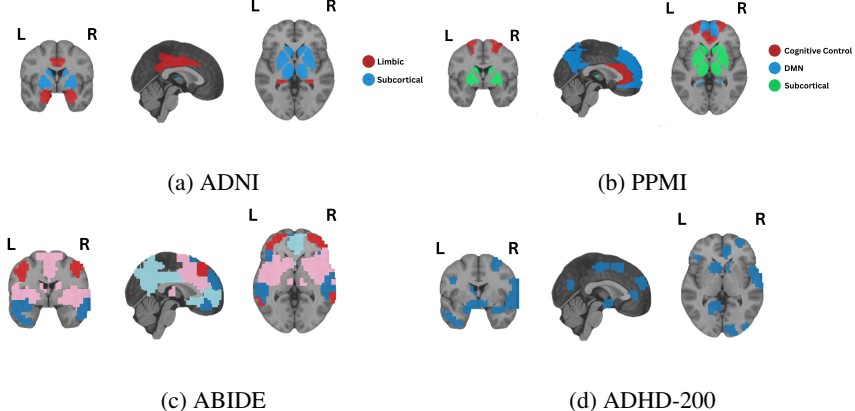

(a) ADNI         (b) PPMI

(c) ABIDE         (d) ADHD-200

Figure 3: Lesion visualization for four diseases. Colored regions indicate highly disease-relevant regions.

### 4.3 ABLATION STUDY

**Effectiveness of Main Modules.** Table 3 summarizes the ablation results (ACC and number of parameter) for the two main modules. F1 and AUC results are shown in Table 8, and results of running time and memory usage are provided in Table 9 (see Appendix F.5). Removing pathological feature refinement causes large, consistent accuracy drops of 36.12%, 39.23%, 34.99%, and 29.41% on ADNI, PPMI, ABIDE, and ADHD-200, respectively, indicating that refinement preserves disease-specific features and prevents noisy node features from misleading the classifier. Removing pathological pattern filter while retaining feature refinement increases parameter counts and produces mixed effects on accuracy. Removing both modules produces the lowest accuracy despite larger model size, confirming that the two-stage approach filters out noise and enhances disease-relevant representations. To further validate the effectiveness of noise feature dropping and pathological feature augmentation in our pathological feature refinement module, we conduct an ablation study by individually removing each component. The results are shown in Table 11 (Appendix F.5).

Table 3: Model performance (ACC (%)) and efficiency (parameter number) of BrainPoG and its variants on four datasets. Pattern filter and feature refinement indicate pathological pattern filter and pathological feature refinement module, respectively.

| Method | ADNI | | PPMI | | ABIDE | | ADHD-200 | |
|---|---|---|---|---|---|---|---|---|
| | ACC | Parameter No. | ACC | Parameter No. | ACC | Parameter No. | ACC | Parameter No. |
| w/o Pattern Filter | 69.86±16.31 | 1.0M | 82.38±2.59 | 253K | 65.20±17.42 | 5.1M | **96.07±3.45** | 2.2M |
| w/o Feature Refinement | 47.19±3.16 | 240K | 43.67±1.02 | **11K** | 58.17±4.70 | 1.7M | 63.62±3.73 | 160K |
| w/o Pattern Filter & Feature Refinement | 44.47±1.61 | 1.0M | 41.07±5.49 | 259K | 52.33±3.14 | 5.1M | 64.95±6.63 | 2.3M |
| **BrainPoG** | **83.31±4.90** | 227K | **82.90±1.62** | 11K | **93.16±2.46** | 415K | 93.03±3.27 | 140K |

**Comparison between Different Dimensionality Reduction Methods.** To compare our pathological feature refinement method with other dimensionality reduction methods, we replace it with (1) PCA (Wold et al., 1987), (2) SVD (Stewart, 1993), (3) Random feature selection (RFS) (Stoppiglia et al., 2003), and (4) Autoencoder (Bank et al., 2023), respectively. Figure 4 compares the performance of models using different dimensionality reduction methods and demonstrates that BrianPoG consistently achieves better performance than other four methods across all four datasets. To assess computational overhead, Table 10 in Appendix F.5 reports parameter counts, per epoch running time and memory usage. Although the other methods demonstrate comparable model efficiency to BrainPoG, their performance deliver inferior performance.

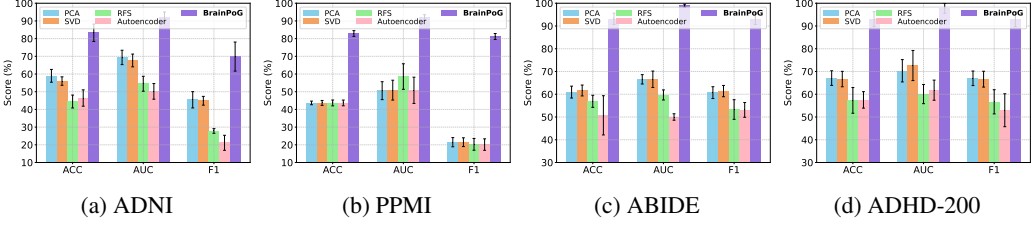

(a) ADNI       (b) PPMI       (c) ABIDE       (d) ADHD-200

Figure 4: Performance of models using different dimensionality reduction methods.

**Effectiveness of Community Detection Methods.** We use spectral clustering as the community detection method to generate subgraphs for the ABIDE and ADHD-200 datasets. To verify its effectiveness, we replace it with three alternatives, including Louvain, density peak clustering, and modularity optimization. As shown in Figure 5, model using spectral clustering yields the best performance on ABIDE and comparable results on ADHD-200. Notably, although it does not achieve the best performance on ADHD-200, its performance across two datasets is more stable compared to other methods.

### 4.4 HYPERPARAMETER STUDY

Figure 5: Performance of models using different community detection methods.

**Impact of $k$ Value.** Noise feature dropping is performed by first computing the communal feature score matrix $\mathbf{S}$, followed by the removal of the top-$k$ and bottom-$k$ ranked features. To explore the impact of different $k$ values to model performance, we conduct experiments with setting $k$ to be 1, 2, 3, and 4. As shown in Figure 6, the model performance remains stable across all $k$ values on all four datasets.

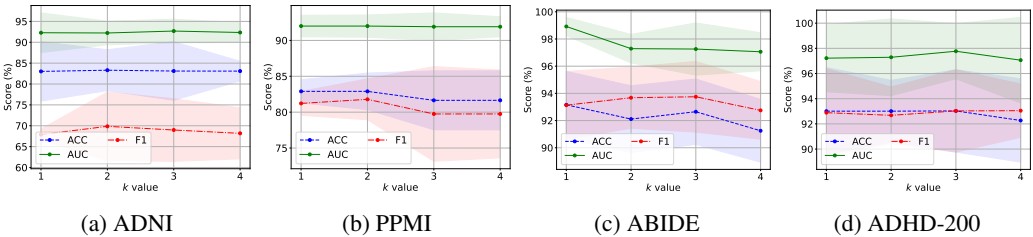

(a) ADNI      (b) PPMI      (c) ABIDE      (d) ADHD-200

Figure 6: Performance of models having different $k$ values.

**Impact of the Magnitude of Feature Enhancement ($\rho$).** Figure 7 illustrates the model performance under different values of $\rho$. As the magnitude of feature enhancement increases, the model performance consistently improves across all four datasets. This is because stronger group-level feature enhancement makes it easier for the model to distinguish between different groups. These results highlight that our pathological feature augmentation method, particularly the group-specific feature importance guided augmentation strategy, plays a crucial role in enhancing the discriminative capability of the model for disease detection.

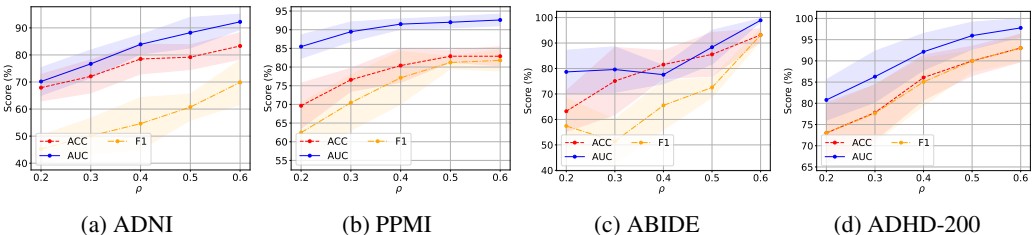

(a) ADNI      (b) PPMI      (c) ABIDE      (d) ADHD-200

Figure 7: Model performance under different $\rho$ values.

## 5 CONCLUSION

This paper presents a lightweight BrainPoG model with a pathological pattern filter and a pathological feature refinement module. Significantly, BrainPoG can focus exclusively on disease-related knowledge while avoiding irrelevant information, enabling efficient brain graph learning. Extensive experiments on four benchmark datasets demonstrate BrainPoG exhibits superiority in both model performance and computational efficiency. This paper contributes to the intersection of neuroscience and artificial intelligence by advancing efficient brain graph learning and enhancing the practicality of models for real-world clinical applications. Although the results are encouraging, the sensitivity of BrainPoG to hyperparameters (e.g., GCN layer number and neuron size) remains a limitation. Therefore, it is worth exploring hyperparameter optimization and more robust architectures to further improve model stability.

ETHICS STATEMENT

This work adheres to the ICLR Code of Ethics. This work uses publicly available neuroimaging datasets (ADNI, PPMI, ABIDE, and ADHD-200), all of which were collected under established ethical guidelines with informed consent from participants and approval from relevant institutional review boards. We strictly followed the data usage agreements and ensured that no personally identifiable information is included in our analysis. Our study focuses on methodological contributions in brain graph learning. The potential societal benefit is to advance reliable tools for diagnosis of neurological diseases. However, we acknowledge that any automated system in healthcare carries risks of misuse or over-interpretation. To mitigate such risks, we emphasize that our method should be regarded as a research tool, not a clinical diagnostic system. Further clinical validation is required before deployment. We declare that there are no conflicts of interest, sponsorship biases, or ethical concerns regarding data privacy, fairness, or security.

REPRODUCIBILITY STATEMENT

We have made the best effort to ensure reproducibility. We have provided a link to an anonymous downloadable source code. Detailed experimental setup has been described in Appendix E.

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

APPENDIX

## A   THE USE OF LARGE LANGUAGE MODELS

Large language models (LLMs) are solely used to aid and polish writing in this manuscript. No data, results, analysis or conclusions are generated by LLMs.

## B   LABELS FOR AAL ATLAS-BASED BRAIN GRAPHS

Table 4: The labels of ROIs and subgraphs of brain graphs constructed based on AAL atlas.

| Subgraph | ROIs |
|---|---|
| Visual | 'Calcarine_L', 'Calcarine_R', 'Cuneus_L', 'Cuneus_R', 'Lingual_L', 'Lingual_R', 'Occipital_Sup_L', 'Occipital_Sup_R', 'Occipital_Mid_L', 'Occipital_Mid_R', 'Occipital_Inf_L', 'Occipital_Inf_R' |
| Auditory | 'Heschl_L', 'Heschl_R', 'Temporal_Sup_L', 'Temporal_Sup_R', 'Temporal_Pole_Sup_L', 'Temporal_Pole_Sup_R' |
| Cognitive Control | 'Frontal_Sup_L', 'Frontal_Sup_R', 'Cingulum_Ant_L', 'Cingulum_Ant_R' |
| Motor | 'Precentral_L', 'Precentral_R', 'Postcentral_L', 'Postcentral_R', 'Rolandic_Oper_L', 'Rolandic_Oper_R', 'Supp_Motor_Area_L', 'Supp_Motor_Area_R' |
| Limbic | 'Hippocampus_L', 'Hippocampus_R', 'Amygdala_L', 'Amygdala_R', 'ParaHippocampal_L', 'ParaHippocampal_R', 'Cingulum_Mid_L', 'Cingulum_Mid_R', 'Cingulum_Post_L', 'Cingulum_Post_R' |
| DMN | 'Frontal_Sup_Medial_L', 'Frontal_Sup_Medial_R', 'Frontal_Med_Orb_L', 'Frontal_Med_Orb_R', 'Precuneus_L', 'Precuneus_R', 'Angular_L', 'Angular_R', 'Cingulum_Post_L', 'Cingulum_Post_R' |
| Sensory | 'Postcentral_L', 'Postcentral_R', 'Parietal_Sup_L', 'Parietal_Sup_R', 'Parietal_Inf_L', 'Parietal_Inf_R' |
| Subcortical | 'Caudate_L', 'Caudate_R', 'Putamen_L', 'Putamen_R', 'Pallidum_L', 'Pallidum_R', 'Thalamus_L', 'Thalamus_R' |
| Olfactory | 'Olfactory_L', 'Olfactory_R' |

## C   PROCEDURE

---

**Algorithm 1** Pathological Pattern Filter

---

1: **Input:** Brain graph dataset $\mathcal{B} = \{\mathcal{G}^{(1)}, \mathcal{G}^{(2)}, ..., \mathcal{G}^{(D)}\}$, where each $\mathcal{G}^{(d)}$ is divided into $m$ subgraphs $\{\mathcal{G}_1^{(d)}, \mathcal{G}_2^{(d)}, ..., \mathcal{G}_m^{(d)}\}$, $D$ is the number of samples.

2: **Output:** PathoGraph set $\hat{\mathcal{B}}=\{\hat{\mathcal{G}}^{(1)}, \hat{\mathcal{G}}^{(2)}, ..., \hat{\mathcal{G}}^{(D)}\}$, where PathoGraph $\hat{\mathcal{G}}^{(d)} = \mathcal{G}^{(d)} - \{\mathcal{G}_i^{(d)}\}_{i=1}^{\hat{m}}$, $\hat{m}$ is the number of dropped subgraphs.

3: Train the SVM with RBF kernel on complete graphs in $\mathcal{B}$.

4: Obtain classification score $\alpha$ for complete graphs in $\mathcal{B}$.

5: Let $\hat{\mathcal{G}} = \mathcal{G}$.

6: **for** subgraph $\mathcal{G}_i$ in $\mathcal{G}$ **do**

7:    Construct subgraph dataset $\mathcal{B}_i = \{\mathcal{G}_i^{(1)}, \mathcal{G}_i^{(2)}, ..., \mathcal{G}_i^{(D)}\}$.

8:    Input $\mathcal{B}_i$ into the SVM and obtain patho-score $\beta_i$.

9:    **if** $\beta_i < \alpha$ **then**

10:       $\hat{\mathcal{G}} = \hat{\mathcal{G}} - \mathcal{G}_i$.

11:    **end if**

12: **end for**

13: **Return** $\hat{\mathcal{B}} = \{\hat{\mathcal{G}}^{(1)}, \hat{\mathcal{G}}^{(2)}, ..., \hat{\mathcal{G}}^{(D)}\}$

---

---

**Algorithm 2** Pathological Feature Refinement

---

1: **Input:** PathoGraph dataset $\hat{\mathcal{B}} = \{\hat{\mathbf{X}}^{(d)} \in \mathbb{R}^{\hat{N} \times \hat{N}}\}_{d=1}^{D}$, which can be divided into $Y$ groups.

2: **Output:** Distilled and enhanced node representations $\{\widetilde{\mathbf{X}}'^{(d)}\}_{d=1}^{D}$.

3: **Step 1: Noise Feature Dropping**

4: **for** $i = 1$ to $\hat{N}$ **do**

5:     Construct cross-subject feature matrix $\mathbf{C}_{v_i} \in \mathbb{R}^{\hat{N} \times D}$ by stacking node $v_i$ across all $D$ subjects.

6:     Compute SVD: $\mathbf{C}_{v_i} = \mathbf{L}_{v_i} \mathbf{\Sigma}_{v_i} \mathbf{R}_{v_i}^{\top}$.

7:     Let $\mathbf{l}_{v_i} = \mathbf{L}_{v_i}[:, 1] = [\mu_{v_i}^1, \ldots, \mu_{v_i}^{\hat{N}}]^{\top}$.

8:     Update communal feature scoring matrix $\mathbf{S}[i, :] \leftarrow \mathbf{l}_{v_i}^{\top}$.

9: **end for**

10: **for** $i = 1$ to $\hat{N}$ **do**

11:     Obtain top-$k$ and bottom-$k$ features for node $v_i$: $(|\mathbf{S}[i, :]|, \text{top-}k)$; $(|\mathbf{S}[i, :]|, \text{bottom-}k)$.

12:     Drop top-$k$ and bottom-$k$ features.

13: **end for**

14: Result: $\widetilde{\mathbf{X}}^{(d)} \in \mathbb{R}^{\hat{N} \times \hat{N}'}$, where $\hat{N}' = \hat{N} - 2k$.

15: **Step 2: Pathological Feature Augmentation**

16: **for** $y = 1$ to $Y$ **do**

17:     Let $D_y = \{d \mid y^{(d)} = y\}$.

18:     **for** $i = 1$ to $\hat{N}$ **do**

19:         Construct group-wise matrix $\mathbf{C}_{v_i}^{(y)} \in \mathbb{R}^{\hat{N}' \times |D_y|}$ from $\{\widetilde{\mathbf{x}}_i^{(d)}\}_{d \in D_y}$.

20:         Compute SVD: $\mathbf{C}_{v_i}^{(y)} = \mathbf{L}_{v_i}^{(y)} \mathbf{\Sigma}_{v_i}^{(y)} \mathbf{R}_{v_i}^{(y)\top}$.

21:         Get group-specific feature scoring vector $\mathbf{F}_{y,i} \leftarrow \mathbf{L}_{v_i}^{(y)}[:, 1]$.

22:     **end for**

23:     Assemble $\mathbf{F}_y \in \mathbb{R}^{\hat{N} \times \hat{N}'}$ with rows $\mathbf{F}_{y,i}^{\top}$.

24:     Compute feature weights $\mathbf{W} \leftarrow \frac{1}{\hat{N}} \sum_{i=1}^{\hat{N}} |\widetilde{\mathbf{x}}_i^{(d)}| \odot \mathbf{F}_{y,i}$ for any $d \in D_y$.

25:     **for** $j = 1$ to $\hat{N}'$ **do**

26:         $p_j \leftarrow \min\left(\rho \cdot \frac{\log w_{max} - \log w_j}{\log w_{max} - \log \lambda_w}, p_t\right)$.

27:         Sample $\tilde{b}_j \sim \text{Bernoulli}(1 - p_j)$.

28:     **end for**

29:     Form group mask $\tilde{\mathbf{b}} \in \{0, 1\}^{\hat{N}'}$.

30:     **for all** $d \in D_y$ **do**

31:         **for** $i = 1$ to $\hat{N}$ **do**

32:             $\widetilde{\mathbf{x}}_i'^{(d)} \leftarrow \widetilde{\mathbf{x}}_i^{(d)} \odot \tilde{\mathbf{b}}$.

33:         **end for**

34:         Stack rows to obtain $\widetilde{\mathbf{X}}'^{(d)}$.

35:     **end for**

36: **end for**

37: **Return:** $\{\widetilde{\mathbf{X}}'^{(d)}\}_{d=1}^{D}$.

---

# D  PROOF OF PROPOSITION 1

**Proposition 1** (Communal features). Based on feature scores, the top ranked features in the cross-subject feature representation $\mathbf{C}_{v_i}$ are considered as communal features of node $v_i$ across all subjects.

*Proof.* For the cross-subject feature matrix $\mathbf{C}_{v_i} \in \mathbb{R}^{\hat{N} \times D}$ of node $v_i$, its SVD is

$$\mathbf{C}_{v_i} = \mathbf{L}_{v_i} \boldsymbol{\Sigma}_{v_i} \mathbf{R}_{v_i}^{\top}, \quad \boldsymbol{\Sigma}_{v_i} = \mathrm{diag}(\sigma_1, \sigma_2, \ldots), \ \sigma_1 \geq \sigma_2 \geq \cdots \geq 0.$$

Let the first left singular vector be

$$\mathbf{l}_{v_i} = \mathbf{L}_{v_i}[:, 1] = \left[ \mu_{v_i}^1, \mu_{v_i}^2, \ldots, \mu_{v_i}^{\hat{N}} \right]^{\top}.$$

By the variational characterization of singular values,

$$\sigma_1 = \max_{\|\mathbf{w}\|_2 = \|\mathbf{z}\|_2 = 1} \mathbf{w}^{\top} \mathbf{C}_{v_i} \mathbf{z},$$

with the maximizer attained at $\mathbf{w} = \mathbf{l}_{v_i}$ and $\mathbf{z} = \mathbf{r}_{v_i}$, where $\mathbf{r}_{v_i}$ is the first right singular vector. Therefore, $\mathbf{l}_{v_i}$ is the feature-space direction that captures the maximum across-subject shared variation.

Moreover, by the Eckart–Young theorem,

$$\arg \min_{\mathrm{rank}(\mathbf{M})=1} \|\mathbf{C}_{v_i} - \mathbf{M}\|_F = \sigma_1 \mathbf{l}_{v_i} \mathbf{r}_{v_i}^{\top},$$

where $\mathbf{M}$ ranges over all rank-one matrices. Hence, the best rank-one feature of $\mathbf{C}_{v_i}$ is given by $\sigma_1 \mathbf{l}_{v_i} \mathbf{r}_{v_i}^{\top}$, and the contribution of each feature dimension to this feature is encoded in the entries of $\mathbf{l}_{v_i}$. Consequently, the score $\mu_{v_i}^j$ quantifies how strongly the $j$-th feature contributes to the dominant across-subject shared component. Ranking features by $\mu_{v_i}^j$ therefore identifies those with the highest shared contribution across subjects, which are considered as the communal features.

$\square$

# E  IMPLEMENTATION DETAILS

The detailed hyperparameter settings for training BrainPoG on four datasets are summarized in Table 5. The model parameters are trained using the Adam optimizer. For the ADNI and PPMI datasets, subgraphs are divided according to the provided labels. Therefore, the community number is not applicable and is denoted as "/".

Table 5: Hyperparameters for training on four different datasets.

| Hyperparameter | ADNI | PPMI | ABIDE | ADHD-200 |
|---|---|---|---|---|
| Noise dropping level ($k$) | 2 | 1 | 1 | 3 |
| Feature enhancement magnitude ($\rho$) | 0.6 | 0.5 | 0.6 | 0.6 |
| Community number | / | / | 7 | 5 |
| #Layers | 4 | 2 | 2 | 2 |
| #Neurons | 128 | 32 | 128 | 64 |
| Dropout | 0.6 | 0.6 | 0.6 | 0.6 |
| Learning rate | 5e-3 | 5e-3 | 5e-3 | 5e-3 |
| #Epochs | 200 | 200 | 200 | 200 |
| Weight decay | 3e-3 | 3e-3 | 3e-3 | 3e-3 |
| Regularization of SVM ($C$) | 1.0 | 1.0 | 1.0 | 1.0 |
| Kernel width of SVM ($\gamma$) | scale | scale | scale | scale |

# F  ADDITIONAL EXPERIMENTAL RESULTS

## F.1  STATISTICAL SIGNIFICANCE ANALYSIS

To verify statistical significance, We conduct two-sample t-tests between BrainPoG and baselines (5 runs per model). Table 6 gives the resulting $p$-values between BrainPoG and all baselines on four datasets. As shown, BrainPoG significantly outperforms almost all baselines ($p < 0.05$) on four datasets.

Table 6: Two-sample t-test $p$-values (ACC, 5 runs) between BrainPoG and all baselines on four datasets.

| Method | ADNI | PPMI | ABIDE | ADHD-200 |
|---|---|---|---|---|
| SVM | 3.6e-05 | 1.4e-05 | 2.2e-05 | 3.6e-05 |
| RF | 1.2e-04 | 2.0e-04 | 6.9e-05 | 1.7e-04 |
| GCN | 2.0e-05 | 1.0e-02 | 2.0e-04 | 2.6e-04 |
| GAT | 2.3e-05 | 4.6e-03 | 8.4e-05 | 7.9e-05 |
| BrainGB | 4.9e-05 | 1.0e-03 | 1.6e-04 | 1.3e-04 |
| BRAINNETTF | 9.0e-04 | 0.115 | 4.0e-04 | 7.0e-04 |
| A-GCL | 1.1e-04 | 1.2e-03 | 1.5e-04 | 2.2e-04 |
| MCST-GCN | 1.4e-02 | 0.162 | 1.3e-03 | 2.2e-03 |
| ALTER | 3.7e-03 | 1.6e-02 | 5.0e-04 | 6.0e-03 |
| BioBGT | 3.0e-04 | 8.0e-03 | 8.0e-04 | 2.0e-03 |
| BrainOOD | 2.0e-03 | 4.0e-02 | 2.8e-03 | 1.5e-02 |
| HGST | 4.1e-04 | 1.4e-04 | 1.5e-02 | 1.4e-04 |

## F.2  INTERPRETABILITY ANALYSIS

We apply the SHAP model to evaluate the interpretability of the proposed method. By computing the SHAP values of the learned brain graph representations, we identified the brain ROIs most influential in brain disease. Notably, since the ABIDE and ADHD-200 datasets do not provide ROI-level labels, the interpretability analysis is conducted only on the ADNI and PPMI datasets. Figure 8 illustrate the top-5 brain ROIs with the highest SHAP values identified in the ADNI dataset, including the hippocampus, caudate, parahippocampal, and pallidum. These ROIs exhibit strong associations with ADNI-related diseases, in particular, pathological alterations in the hippocampus are directly implicated in the development of Alzheimer's disease (Salta et al., 2023). For Parkinson's disease, the thalamus, frontal cortex, and cingulum are identified as the ROIs with the highest SHAP values, as illustrated in Figure 9. The identified ROIs show strong associations with Parkinson's disease predictions, which aligns closely with evidence reported in prior neuroscience studies on Parkinson's disease (Chen et al., 2023; Pagonabarraga et al., 2024). These results demonstrate that BrainPoG effectively identifies disease-relevant brain ROIs, providing interpretable insights that align with established neuroscience knowledge for both Alzheimer's and Parkinson's diseases.

## F.3  TIME AND SPACE COMPLEXITY

To further verify the efficiency of BrainPoG, we compare its per-graph inference complexity with other state-of-the-art brain graph learning models (see Table 7). For all baselines, brain graphs are typically dense and require processing all ROIs, leading to both time and space complexity of order $\mathcal{O}(N^2)$, with constant factors depending on the number of layers, heads, or additional modules. In contrast, BrainPoG performs inference on a filtered PathoGraph of retained subgraphs, which reduces the effective graph size to $\hat{N} \ll N$, resulting in both time and space complexity of $\mathcal{O}(\hat{N}^2)$. Thus, while the asymptotic order remains quadratic, BrainPoG operates on a much smaller graph and achieves substantially lower running time and memory consumption during per-graph inference.

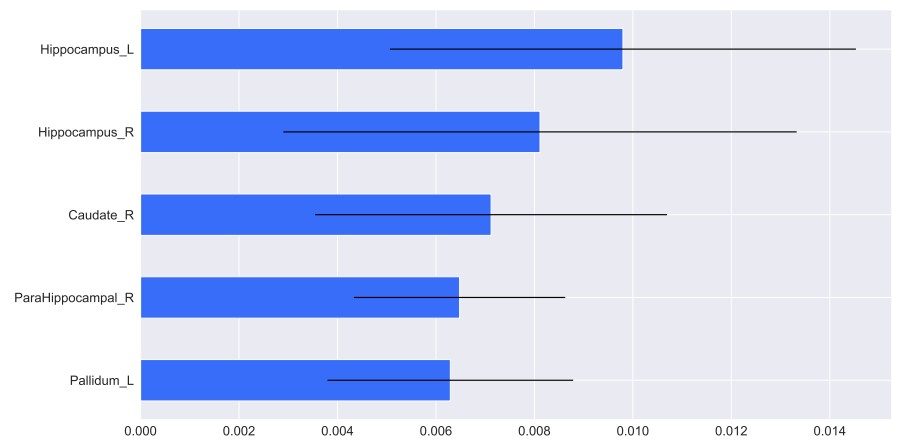

Figure 8: Visualization of the top-5 ROIs with the highest SHAP values for ADNI dataset.

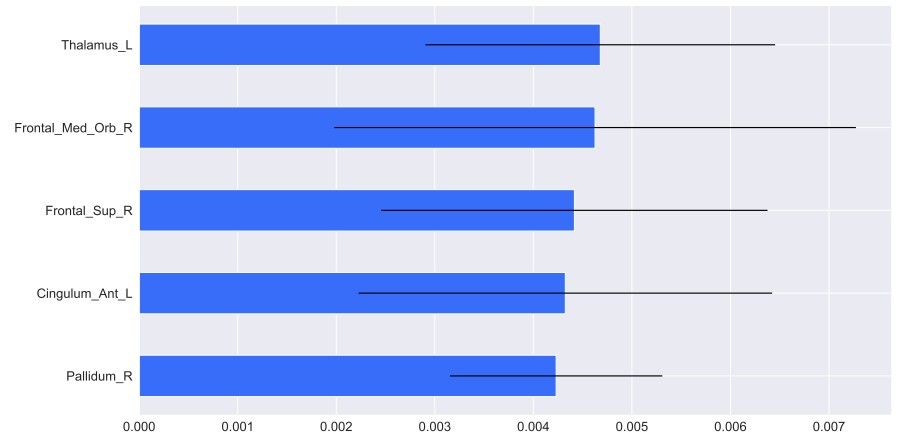

Figure 9: Visualization of the top-5 ROIs with the highest SHAP values for PPMI dataset.

Table 7: Time and space complexity comparison between BrainPoG and state-of-the-art brain graph learning models.

| Method | Time complexity | Space complexity |
|---|---|---|
| BrainGB | $\mathcal{O}_t(N^2)$ | $\mathcal{O}_s(N^2)$ |
| BRAINNETTF | $\mathcal{O}_t(N^2)$ | $\mathcal{O}_s(N^2)$ |
| A-GCL | $\mathcal{O}_t(N^2)$ | $\mathcal{O}_s(N^2)$ |
| MCST-GCN | $\mathcal{O}_t(N^2)$ | $\mathcal{O}_s(N^2)$ |
| ALTER | $\mathcal{O}_t(N^2)$ | $\mathcal{O}_s(N^2)$ |
| BioBGT | $\mathcal{O}_t(N^2)$ | $\mathcal{O}_s(N^2)$ |
| BrainOOD | $\mathcal{O}_t(N^2)$ | $\mathcal{O}_s(N^2)$ |
| **BrainPoG** | $\mathcal{O}_t(\hat{N}^2)$ | $\mathcal{O}_s(\hat{N}^2)$ |

## F.4 PATHOGRAPH VISUALIZATION

Figures 10 and 11 present group-wise heatmaps of PathoGraph representations (adjacency matrices of PathoGraphs) for the ADNI and PPMI test sets, respectively. Notably, because brain graphs in ABIDE and ADHD-200 lack ROI labels, it is not feasible to obtain the correlation patterns of specific subgraphs from their PathoGraph visualizations. Therefore, we visualize the PathoGraph representations for the ADNI and PPMI datasets. In both ADNI and PPMI datasets, the representations show significant difference across diagnostic groups (AD, MCI, and NC in ADNI; PD, Prodromal, and NC in PPMI). These results demonstrate BrainPoG can effectively reveal disease-specific alterations.

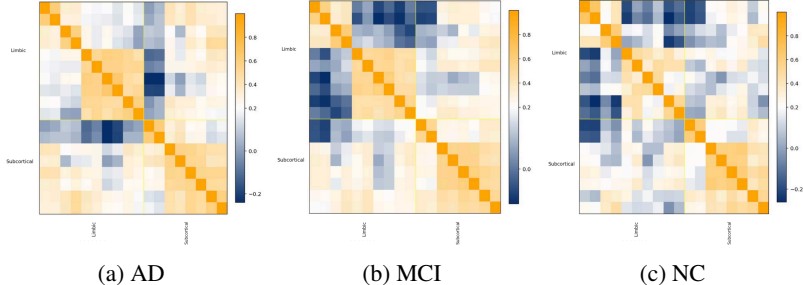

|          (a) AD           |          (b) MCI          |          (c) NC           |

Figure 10: Group-wise heatmaps of PathoGraph representations in ADNI dataset.

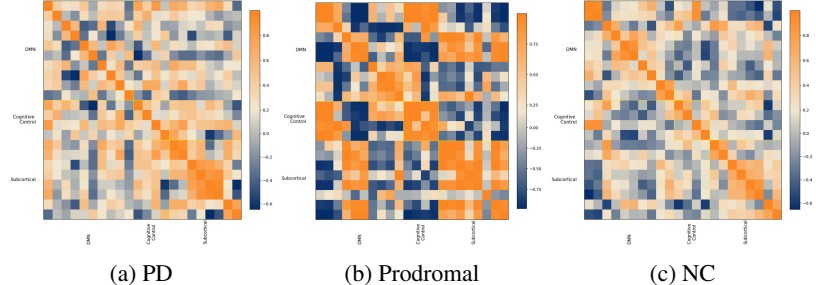

|          (a) PD           |        (b) Prodromal       |          (c) NC           |

Figure 11: Group-wise heatmaps of PathoGraph representations in PPMI dataset.

## F.5 ADDITIONAL RESULTS OF ABLATION STUDY

Tables 8 and 9 respectively give the model performance (AUC and F1) and efficiency (running time and memory) of BrainPoG and its variants on four datasets. Table 10 shows the efficiency of models using different dimensionality reduction methods.

Table 8: Model performance (AUC and F1) of BrainPoG and its variants on four datasets (%).

| Method | ADNI | | PPMI | | ABIDE | | ADHD-200 | |
|---|---|---|---|---|---|---|---|---|
| | AUC | F1 | AUC | F1 | AUC | F1 | AUC | F1 |
| w/o Pattern Filter | 75.26±14.78 | 47.33±15.05 | 92.09±2.33 | 92.08±2.68 | 74.89±17.67 | 56.88±23.04 | 99.81±0.02 | 96.04±3.49 |
| w/o Feature Refinement | 54.49±4.47 | 29.66±2.67 | 60.77±11.55 | 21.62±2.76 | 66.04±6.04 | 50.10±9.77 | 71.63±5.48 | 62.47±3.66 |
| w/o Pattern Filter & Feature Refinement | 55.65±5.40 | 23.85±2.52 | 58.01±4.90 | 24.46± 5.55 | 53.96±4.94 | 41.93±10.49 | 73.64±5.96 | 64.52±7.04 |
| **BrainPoG** | 92.23±2.83 | 69.86±8.21 | 92.00±1.50 | 81.22±1.66 | 98.92±0.67 | 93.13±2.49 | 97.77±2.19 | 93.03±3.27 |

Table 9: Model efficiency (running time and memory usage) of BrainPoG and its variants on four datasets.

| Method | ADNI | | PPMI | | ABIDE | | ADHD-200 | |
|---|---|---|---|---|---|---|---|---|
| | Runing time | Memory | Runing time | Memory | Runing time | Memory | Runing time | Memory |
| w/o Pattern Filter | 0.0047s/epoch | 48MB | 0.0033s/epoch | 26MB | 0.0041s/epoch | 269MB | 0.0031s/epoch | 121MB |
| w/o Feature Refinement | 0.0051s/epoch | 59MB | 0.0029s/epoch | 21MB | 0.0044s/epoch | 237MB | 0.0029s/epoch | 28MB |
| w/o Pattern Filter & Feature Refinement | 0.0055s/epoch | 110MB | 0.0029s/epoch | 27MB | 0.0051s/epoch | 325MB | 0.0031s/epoch | 124MB |
| **BrainPoG** | 0.0046s/epoch | 23MB | 0.0029s/epoch | 17MB | 0.0030s/epoch | 99MB | 0.0030s/epoch | 23MB |

Table 10: Efficiency of models using different dimensionality reduction methods.

| Method | ADNI | | | PPMI | | | ABIDE | | | ADHD-200 | | |
|---|---|---|---|---|---|---|---|---|---|---|---|---|
| | Parameter No. | Running time | Memory | Parameter No. | Running time | Memory | Parameter No. | Running time | Memory | Parameter No. | Running time | Memory |
| PCA | 34K | 0.0030s/epoch | 17MB | 17K | 0.0031s/epoch | 19MB | 68K | 0.0027s/epoch | 25MB | 33K | 0.0030s/epoch | 17MB |
| SVD | 39K | 0.0047s/epoch | 19MB | 17K | 0.0030s/epoch | 19MB | 67K | 0.0029s/epoch | 25MB | 33K | 0.0030s/epoch | 17MB |
| RFS | 40K | 0.0052s/epoch | 53MB | 17K | 0.0029s/epoch | 20MB | 67K | 0.0046s/epoch | 26MB | 33K | 0.0029s/epoch | 35MB |
| Autoencoder | 40K | 0.0061s/epoch | 127MB | 17K | 0.0031s/epoch | 22MB | 67K | 0.0071s/epoch | 526MB | 33K | 0.0029s/epoch | 98MB |
| **BrainPoG** | 227K | 0.0046s/epoch | 23MB | 11K | 0.0029s/epoch | 17MB | 415K | 0.0030s/epoch | 99MB | 140K | 0.0030s/epoch | 23MB |

**Effectiveness of Two Components in Pathological Feature Refinement Module.** To validate the effectiveness of noise-feature dropping and pathological feature augmentation in our pathological feature refinement module, we perform an ablation study in which each component is removed in turn. The results are reported in Table 11. '*w/o Noise Dropping*' denotes the model without the noise-feature dropping component, and '*w/o Feature Augmentation*' indicates the model without the pathological feature augmentation component. Across all four datasets, removing pathological feature augmentation yields a marked performance drop, indicating that this strategy effectively enhances group-discriminative representations by amplifying group-specific features. Omitting noise-feature dropping likewise degrades accuracy and worsens efficiency, underscoring its role in reducing redundancy and preventing the model from learning disease-irrelevant features.

Table 11: Model performance (ACC (%)) and efficiency (parameter number) of BrainPoG and its variants on four datasets.

| Method | ADNI | | PPMI | | ABIDE | | ADHD-200 | |
|---|---|---|---|---|---|---|---|---|
| | ACC | Parameter No. | ACC | Parameter No. | ACC | Parameter No. | ACC | Parameter No. |
| w/o Noise Dropping | 78.65±4.44 | 257K | 87.34±2.05 | 65K | 73.35±14.72 | 1.7M | 88.24±2.76 | 164K |
| w/o Feature Augmentation | 45.72±5.71 | 245K | 45.50±4.91 | 63K | 54.51±2.77 | 1.6M | 59.68±5.39 | 145K |
| **BrainPoG** | 83.31±4.90 | 227K | 82.90±1.62 | 11K | 93.16±2.46 | 415K | 93.03±3.27 | 140K |

## F.6 ADDITIONAL RESULTS OF HYPERPARAMETER STUDY

**Experimental Results under Different Numbers of Layers and Neurons.** To investigate model performance under different configurations and identify the optimal setting, we conduct experiments with varying numbers of GCN layers and neurons. The results are presented in Figure 12.

**Model Performance under Different Community Numbers.** As mentioned before, we use spectral clustering to obtain subgraphs for brain graphs in ABIDE and ADHD-200 datasets. As spectral clustering is unsupervised method, different community numbers can be defined. To investigate the influence of community number on model performance, we conduct experiments with different community numbers. Figure 13 shows the model performance under different community numbers. As shown, the community number has a substantial impact on the model performance. This indicates the importance of subgraph division, reflecting functional modules.

**PathoGraph Identification under Different SVM Settings.** PathoGraph is identified based on the patho-score computed using an RBF-kernel SVM with regularization parameter $C = 1.0$ and kernel width $\gamma = \texttt{scale}$. The SVM is evaluated under five different random seeds $\{1, 11, 111, 112, 113\}$, using a 75%–25% train–test split. To assess the sensitivity of PathoGraph identification to SVM hyperparameters, we conduct experiments under different SVM settings, including varying values of $C$ and $\gamma$, as well as different random seeds and data split ratios. Since

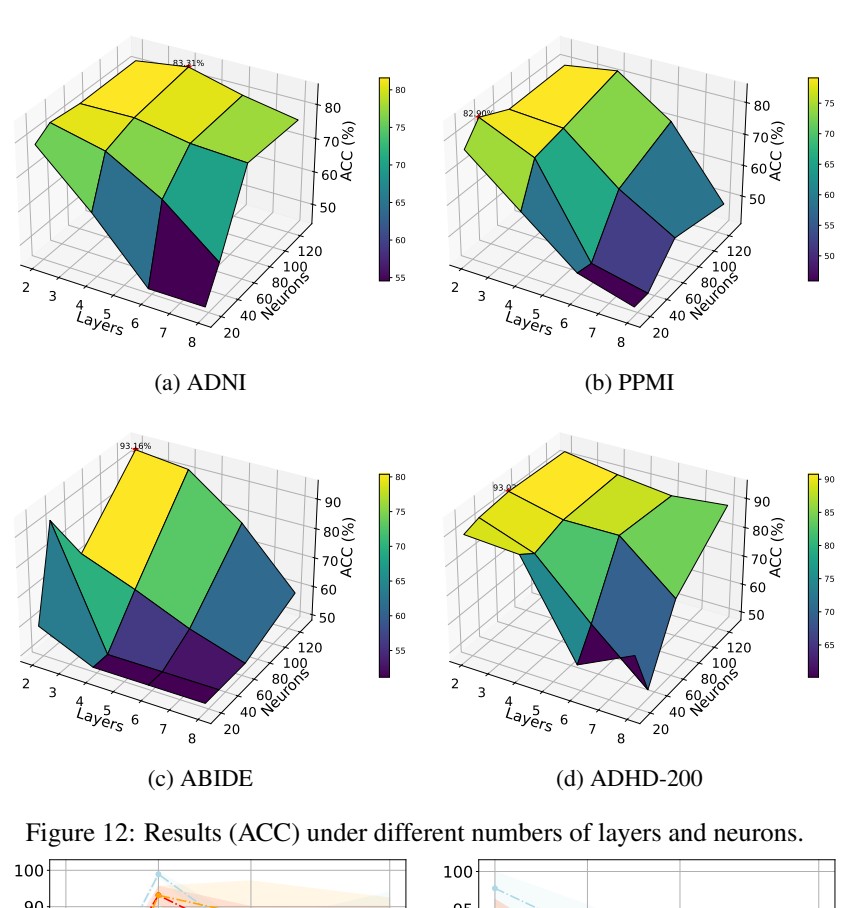

Figure 12: Results (ACC) under different numbers of layers and neurons.

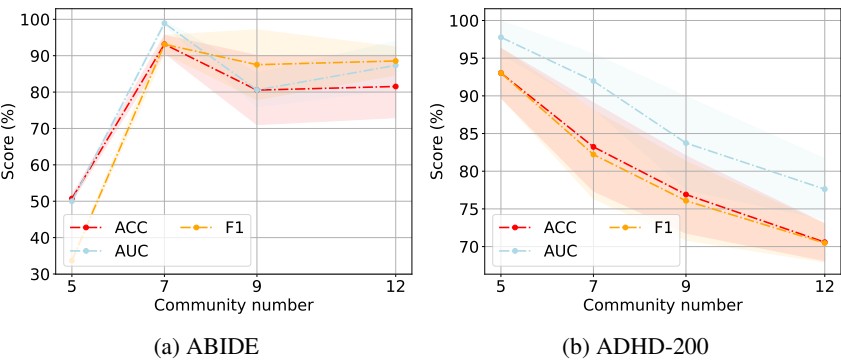

Figure 13: Model performance under different community numbers in ABIDE and ADHD-200 datasets.

ADNI and PPMI provide subgraph labels, they allow intuitive qualitative comparison to evaluate whether the extracted PathoGraphs remain consistent across different SVM configurations. Therefore, we perform this hyperparameter study primarily on ADNI and PPMI datasets.

Firstly, we conduct experiments using different SVM hyperparameters, including $C \in \{1.0, 10.0\}$ and $\gamma \in \{\texttt{scale}, 0.1\}$. The resulting PathoGraphs are shown in Figure 14 for ADNI and Figure 15 for PPMI. As illustrated, the identified PathoGraphs remain highly consistent across all parameter settings. The extracted PathoGraphs are the same under all tested configurations for both datasets, demonstrating the stability of the proposed SVM-based scoring method with respect to hyperparameter variations.

Then, we evaluate the robustness of PathoGraph identification under different random seed sets used for training the SVM. Specifically, we consider three seed sets: $\{1, 11, 111, 112, 113\}$, $\{2, 22, 222, 280, 290\}$, and $\{3, 33, 333, 350, 390\}$. Figures 16 and 17 show the results for ADNI and PPMI, respectively. Overall, the identified PathoGraphs remain consistent across all seed sets.

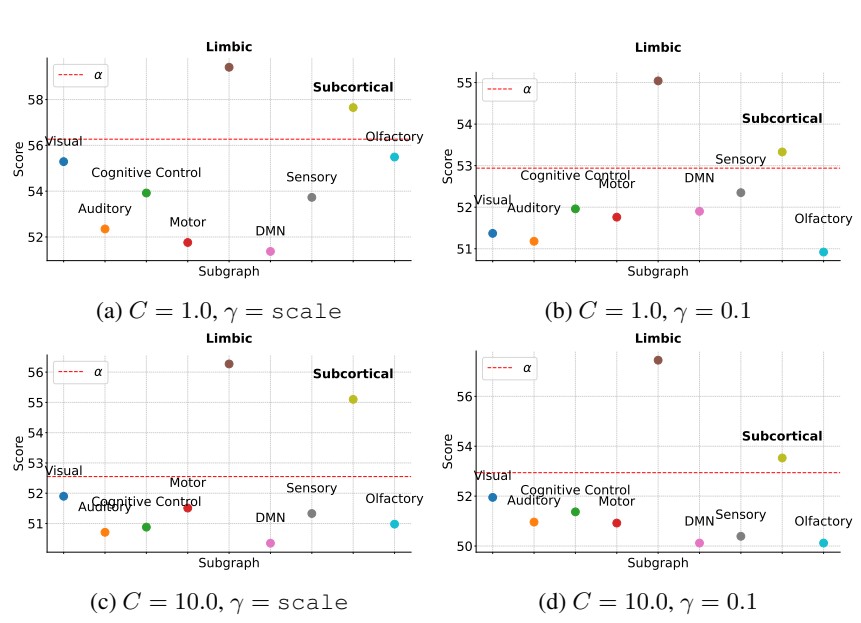

Figure 14: PathoGraph identification under different $C$ and $\gamma$ of SVM on ADNI.

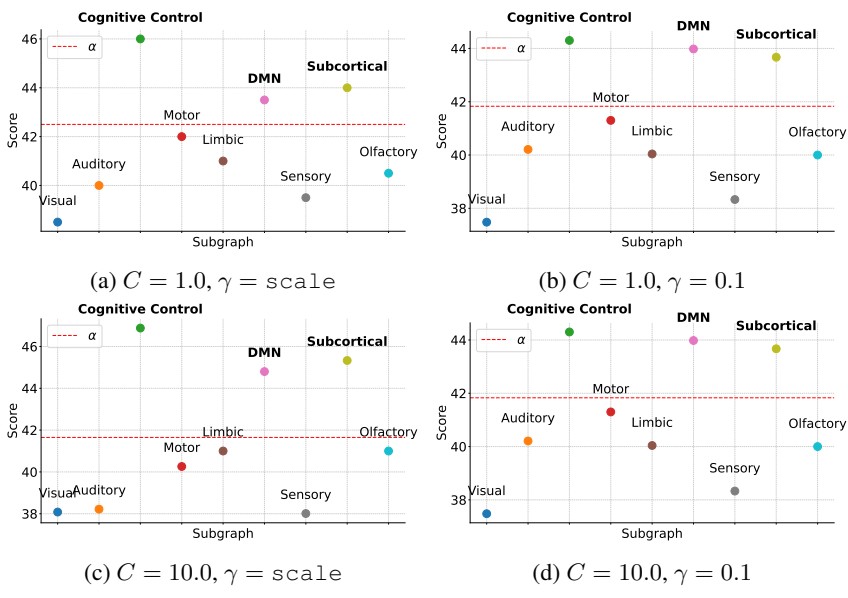

Figure 15: PathoGraph identification under different $C$ and $\gamma$ of SVM on PPMI.

Although the ADNI results under the seed set $\{2, 22, 222, 280, 290\}$ show a slight deviation (e.g., the 'Visual' subgraph is additionally included in the PathoGraph), the overall disease-relevant subgraphs remain highly consistent. For ADNI, the 'Limbic' and 'Subcortical' subgraphs consistently yield patho-scores higher than the classification score $\alpha$. For PPMI, the 'Cognitive Control', 'DMN', and 'Subcortical' subgraphs consistently achieve patho-scores higher than $\alpha$. These results demonstrate that PathoGraph identification is robust to variations in random seeds.

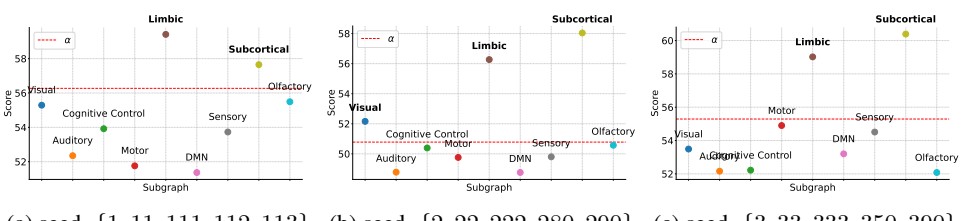

(a) seed=$\{1, 11, 111, 112, 113\}$ (b) seed=$\{2, 22, 222, 280, 290\}$ (c) seed=$\{3, 33, 333, 350, 390\}$

Figure 16: PathoGraph identification under different seed sets of SVM on ADNI.

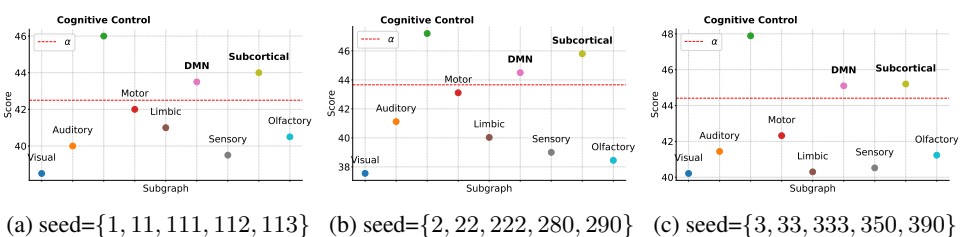

(a) seed=$\{1, 11, 111, 112, 113\}$ (b) seed=$\{2, 22, 222, 280, 290\}$ (c) seed=$\{3, 33, 333, 350, 390\}$

Figure 17: PathoGraph identification under different seed sets of SVM on PPMI.

In addition, we conduct experiments using different train–test split ratios, including 75%–25%, 70%–30%, and 80%–20%. The results for the two datasets are shown in Figure 18 and Figure 19, respectively. As illustrated, the identified PathoGraphs remain almost identical across all data split configurations. The only exception occurs on PPMI under the 80%–20% split, where the 'DMN' subgraph is not included as part of the PathoGraph. Nevertheless, the core disease-relevant subgraphs remain consistent across different splits, demonstrating that PathoGraph identification is generally stable with respect to variations in train–test partitioning.

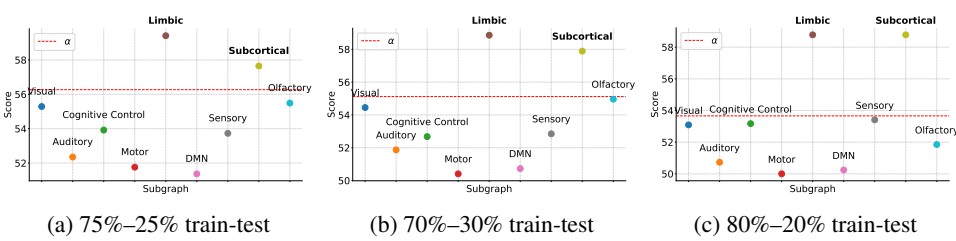

(a) 75%–25% train-test (b) 70%–30% train-test (c) 80%–20% train-test

Figure 18: PathoGraph identification under different data split ratios on ADNI.

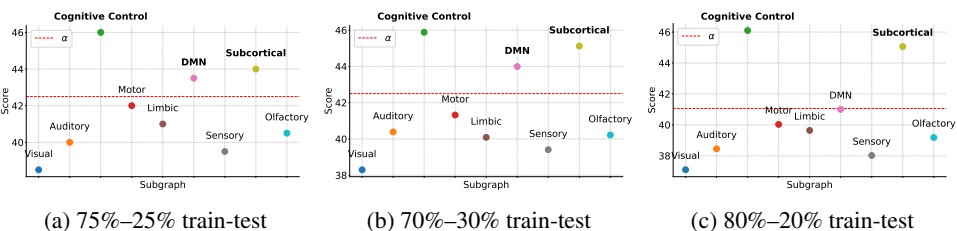

(a) 75%–25% train-test (b) 70%–30% train-test (c) 80%–20% train-test

Figure 19: PathoGraph identification under different data split ratios on PPMI.

**PathoGraph Identification with Different Methods.** To compare the PathoGraph identification obtained from SVM with other classifiers such as RF and GCN, we also performed PathoGraph identification experiments using RF and GCN. Figure 20 and Figure 21 show the results on ADNI and PPMI, respectively. As shown, the identified disease-relevant subgraphs were highly consistent across all three classifiers, indicating that our classifier-based strategy for detecting highly disease-relevant patterns is both reliable and effective. SVM was ultimately chosen due to its lightweight design and low computational cost, making it well suited for repeated subgraph-level evaluations.

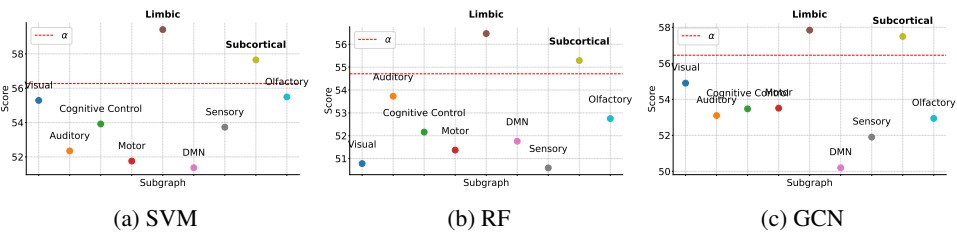

Figure 20: PathoGraph identification using different methods on ADNI.

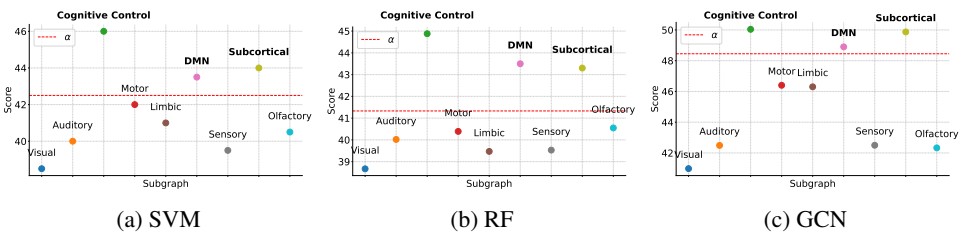

Figure 21: PathoGraph identification using different methods on PPMI.