# OpenReview forum: "Brain PathoGraph Learning"
_ICLR.cc/2026/Conference — Submitted to ICLR 2026_

### Official Review · Reviewer_ZBLs · 2025-10-17

**Soundness:** 1
**Presentation:** 2
**Contribution:** 1
**Rating:** 2
**Confidence:** 5

**Summary:**

This paper proposes a lightweight brain graph learning framework, named Brain PathoGraph Learning (BrainPoG), aiming to improve the efficiency and interpretability of brain disease analysis. The authors argue that existing brain graph methods often fail to focus on disease-related patterns, resulting in high computational costs and redundant parameters. To address this, BrainPoG introduces mechanisms for pathological pattern filtering and pathological feature distillation, enabling selective learning of disease-relevant representations. The paper further provides analyses of disease-related regions to support the interpretability of the extracted features.

**Strengths:**

### **The paper makes a meaningful attempt to extract disease-specific brain features.**
By focusing on identifying brain patterns that are specific to different neurological or psychiatric disorders, the work contributes to improving the interpretability of AI-based brain disease prediction models. This direction is important not only for enhancing diagnostic accuracy but also for advancing our understanding of the underlying neurobiological mechanisms.
### **The introduction of PathoGraph as a disease-specific subgraph representation is conceptually innovative.**
Instead of modeling the whole-brain connectome, the proposed method isolates pathological subgraphs that capture disease-relevant interactions. This approach offers a perspective on selective brain network learning and represents a creative step toward localizing disease-related brain regions in graph-based analysis.
### **The visualization and clinical interpretation of disease-specific brain regions enhance the study’s significance.**
The authors visualize the brain regions identified by their model, providing an intuitive understanding of the disease-specific activation patterns. They further relate these findings to previously reported results in the literature, offering supporting evidence that reinforces the validity and interpretability of the conclusions.

**Weaknesses:**

### **The motivation is not sufficiently convincing; the necessity of model lightweighting needs re-examination.**
The core motivation of reducing model parameters and computational costs seems weak, as most brain network models are already relatively small and can be easily handled with current computational resources. There is currently little demand for deployment in low-resource environments. To make the motivation more persuasive, the authors are encouraged to clearly define a realistic low-resource or deployment scenario (e.g., edge devices, privacy-preserving federated settings, or multi-site low-resource hospitals) where lightweight modeling is genuinely beneficial.
### **The paper relies on a strong and controversial assumption that brain diseases correspond to specific lesion regions or circuits.**
This assumption is debatable, especially for neurodegenerative and psychiatric disorders, which often show distributed and network-level dysfunction rather than focal lesions. Theoretical justification is weak—the introduction cites only two references (Mahesh & Tasneem, 2014; Le et al., 2025), one lacking impact and the other a preprint, which are insufficient to support such a strong claim. The authors could strengthen this part by:
1. expanding the literature review to distinguish between diseases with localized lesions (e.g., structural or tumor-related) and those with diffuse, network-wide alterations (e.g., schizophrenia, depression), and clarifying BrainPoG’s applicability in both contexts;
2. validating the model separately on datasets representing these two categories to assess robustness to different pathophysiological assumptions; and
3. adding hypothesis tests (e.g., random-label or spatial permutation tests) to verify that the extracted “PathoGraphs” capture meaningful disease patterns beyond random or confounding effects.
### **The originality claim is limited; the idea of subgraph extraction has precedents, and related work is not sufficiently compared.**
The proposed PathoGraph shares conceptual similarities with previous work (e.g., Le et al., 2025) that already introduced disease-specific subgraph extraction, and with hypergraph-based approaches for modeling brain networks (e.g., Multiview hyperedge-aware hypergraph embedding learning for multisite, multiatlas fMRI-based functional connectivity network analysis). The lack of direct comparison with these studies weakens the novelty. The authors should:
1. explicitly clarify the methodological and conceptual differences from prior subgraph or hypergraph-based models;
2. include at least one or two of these methods as baselines for quantitative comparison; and
3. provide theoretical or empirical analysis explaining how BrainPoG improves over prior work (e.g., through architectural simplification, interpretability, or robustness).
### **Serious experimental concerns: potential label leakage and over-claimed results undermine credibility.**
The reported over 90% binary classification accuracy on ABIDE and ADHD datasets is surprisingly high and difficult to believe. From Table 3, it appears that the performance gain primarily comes from the Feature Distillation (FD) module, while PathoGraph itself may even reduce performance in some settings. As described, FD uses the entire dataset and label information, introducing a major label leakage risk that compromises result validity. The authors must carefully recheck their data pipeline and address the following issues:
1. Eliminate leakage: Re-implement FD using only the training set and ensure no test data or labels are accessed during distillation or feature extraction.
2. Clarify data splits: Clearly report how data were divided (train/validation/test), whether site stratification was used, and whether any cross-site evaluation was conducted. Ideally, introduce a completely held-out external test set for unbiased evaluation.
3. Adopt stricter evaluation: Use nested cross-validation or fixed train/test splits with confidence intervals and statistical tests (e.g., bootstrapping or random resampling) to verify the significance of performance gains.
4. Report ablations rigorously: Present results for models without FD, without PathoGraph, and with standard baselines. If FD accounts for most improvements, describe its setup in detail (data used, supervision level, teacher model, loss function) and confirm the benefit persists under strict no-leakage conditions.
5. Validate robustness: Conduct stability tests (different random seeds, sub-sampling) and permutation tests to confirm that the detected regions are not artifacts of data bias or preprocessing.

**Addressing these concerns would substantially strengthen the paper. I will reconsider my overall score if the authors can adequately address the concerns raised above.**

**Questions:**

1. Why did the authors use different brain atlases across datasets instead of adopting a consistent atlas such as AAL for both ABIDE and ADHD?
2. Given that the Schaefer atlas provides well-defined 7/17 functional subnetworks, why was it not used for subnetwork-based analysis?
3. How exactly does the SVM accept graph or subgraph representations as input, and what feature transformation or embedding process is applied?
4. If disease-specific feature extraction is beneficial, why were pretrained models that learn generalizable representations not included for comparison?
5. Considering that SVM shows limited predictive performance in the results, why was it chosen as the subgraph selection evaluator?

---

> ### Author Response · Authors · 2025-11-23
> **Response to  Reviewer ZBLs (1)**
>
> Thank you very much for pointing out the issues in our paper and providing your insightful suggestions. Based on your comments, we provide detailed responses below.
>
> ---
>
> **Regarding the motivation**: We appreciate your kind comment. We agree that clarifying a realistic deployment scenario can make the motivation more convincing. The main motivation of this work is to reduce heavy parameters and computational cost in brain graph learning models. Here, we’d like to emphasize that lightweight design is particularly necessary for **privacy-sensitive and low-resource clinical deployment**. Brain graph learning models are often required to be deployed on on-premise clinical workstations or local hospital servers that often have limited GPUs and computational capacity due to strict medical data privacy regulations (e.g., Health Insurance Portability and Accountability Act, HIPAA). In such cases, cloud-based inference is not permitted, and lightweight models enable on-site processing without transmitting sensitive brain data outside the hospital network. This ensure both data privacy and computational efficiency. We have revised the introduction to better highlight these realistic scenarios. Please refer to lines 40-47.
>
> **Concern about disease–region correspondence**: Thank you very much for your insightful comment. We’d like to clarify that we do not assume that brain diseases correspond to specific lesion regions. Instead, we emphasize **region-specific vulnerability** [1], which describe the varying degrees of relevance between specific brain regions and diseases. Particularly, we claim that a certain brain disease is highly relevant to some specific regions and may be less relevant to some regions [1, 2]. While neurodegenerative and psychiatric disorders indeed involve distributed and network-level dysfunctions, many exhibit regionally concentrated alterations [2, 3]. For example, Alzheimer’s disease prominently affects the hippocampus regions in Limbic subgraph [2].
>
> In our experiments, we validated this perspective using the ADNI (Alzheimer’s disease) and PPMI (Parkinson’s disease) datasets. As shown in Fig. 3, Alzheimer’s disease is highly associated with limbic and subcortical subgraphs [4,5], whereas Parkinson’s disease is highly relevant to cognitive control, DMN, and subcortical subgraphs [6,7]. Both of them consistent with established neuroimaging literature. Although we have not absolutely identified all highly relevant regions, our model effectively captures highly disease-relevant regions.
>
> [1] Seeley, William W., et al. "Neurodegenerative diseases target large-scale human brain networks." Neuron (2009).
>
> [2] Grothe, Michel J., et al. "Molecular properties underlying regional vulnerability to Alzheimer’s disease pathology." Brain (2018).
>
> [3] Braak, Heiko, and Eva Braak. "Neuropathological stageing of Alzheimer-related changes." Acta neuropathologica (1991).
>
> [4] Yang, Zhijian, et al. "A deep learning framework identifies dimensional representations of Alzheimer’s Disease from brain structure." Nature communications (2021).
>
> [5] Seoane, Sara, et al. "The subcortical default mode network and Alzheimer’s disease: a systematic review and meta-analysis." Brain Communications (2024).
>
> [6] Aarsland, Dag, et al. "Parkinson disease-associated cognitive impairment." Nature reviews Disease (2021).
>
> [7] Yau, Y., et al. "Network connectivity determines cortical thinning in early Parkinson’s disease progression." Nature communications (2018).

---

> ### Author Response · Authors · 2025-11-23
> **Response to Reviewer ZBLs (2)**
>
> **Regarding the Originality**: Thank you for your comment. We’d like to clarify that our proposed BrainPoG differs conceptually from previous methods. Prior works typically extract subgraphs or construct hypergraphs to represent the whole-brain connectivity structure in hypergraph level. In contrast, our approach explicitly aims to identify and retain disease-relevant subgraphs, selectively learn highly disease-relevant subgraphs (PathoGraph) rather than modeling the entire brain graph. According to your valuable suggestion, we have added a new baseline ([8]) for comparison (see the following table). The results are also added in the revised manuscript (see Table 1).
> In addition, our model introduces architectural simplification over previous works. Specifically, the Pathological Pattern Filter module prunes the brain graph to preserve disease-relevant substructures, while the Pathological Feature Distillation module removes disease-irrelevant noise and enhances pathological features. Together, these designs improve both model efficiency and disease-specific feature learning.
>
> | Method   | ADNI (ACC / AUC / F1)      | PPMI (ACC / AUC / F1)      | ABIDE (ACC / AUC / F1)      | ADHD-200 (ACC / AUC / F1)      |
> |----------|-----------------------------|------------------------------|-------------------------------|--------------------------------|
> | HGST     | 57.33±2.77 / 61.02±3.15 / 54.76±2.80 | 66.24±3.52 / 64.21±3.29 / 63.89±4.02 | 68.89±4.15 / 68.02±2.52 / 66.89±4.06 | 70.91±2.52 / 68.94±3.96 / 69.30±4.14 |
> | BrainPoG | 83.31±4.90 / 92.23±2.83 / 69.86±8.21 | 82.90±1.62 / 92.00±1.50 / 81.22±1.66 | 93.16±2.46 / 98.92±0.67 / 93.13±2.49 | 93.03±3.27 / 97.77±2.19 / 93.03±3.27 |
>
>
> [8] Han, Xiangmin, et al. "Hypergraph-based semantic and topological self-supervised learning for brain disease diagnosis." Pattern Recognition (2026).
>
> **Concern about potential label leakage**: Thank you very much for raising this important concern. In our model, the pathological feature augmentation module uses label information only to guide group-specific feature augmentation, where samples are first divided into groups according to their labels, and feature augmentation is performed within each group separately. After augmentation, the label information is discarded and all samples are mixed and fed into the main training pipeline. Therefore, the labels are not injected into the feature values themselves.
>
> Following your valuable suggestion, we conducted additional experiments to further ensure that the feature augmentation stage does not have label leakage issue:
>
> 1. We replaced label-based grouping with unsupervised spectral clustering, where groups are constructed solely from data without using any labels. The results (ACC) are shown in the following table. Using unsupervised grouping leads to slightly lower performance, as expected, because label-guided group division leads more accurate group-specific feature augmentation. However, the performance remains strong, demonstrating that the module’s effectiveness is not due to label leakage, but due to denoising and feature  augmentation.
>
> | Method                                 | ADNI          | PPMI          | ABIDE         | ADHD-200      |
> |----------------------------------------|---------------|---------------|---------------|---------------|
> | BrainPoG (original, label-guided)      | 83.31±4.90    | 82.90±1.62    | 93.16±2.46    | 93.03±3.27    |
> | BrainPoG (spectral clustering)         | 79.24±4.43    | 78.50±3.67    | 85.96±2.15    | 84.61±2.64    |
>
>
> 2.We use stratified five-fold cross-validation, in each fold, the dataset is randomly partitioned into 70% training, 10% validation, and 20% test set.The test set is completely held-out external. To strengthen evaluation rigor, we report two-sample t-test results across 5 runs for all datasets and all baselines (See Table 6 in the Appendix F.1).
>
> 3.We already provide ablations (see Table 3 and Table 11). The performance gain of FD primarily comes from removing disease-irrelevant noise and enhancing group-specifical features.

---

### Official Review · Reviewer_RNc7 · 2025-10-26

**Soundness:** 2
**Presentation:** 2
**Contribution:** 2
**Rating:** 4
**Confidence:** 5

**Summary:**

This paper proposes BrainPoG, a lightweight brain graph learning framework designed to improve computational efficiency and disease-specific interpretability in fMRI-based brain network analysis. The method consists of two core modules: a pathological pattern filter that constructs a PathoGraph by retaining only highly disease-relevant subgraphs using SVM-based patho-scores, and a pathological feature distillation module that removes noise features and enhances discriminative pathological features via SVD and group-specific masking. Extensive experiments on four public neuroimaging datasets (ADNI, PPMI, ABIDE, ADHD-200) demonstrate consistent improvements in both accuracy and efficiency over state-of-the-art baselines.

While BrainPoG presents a promising and well-motivated framework with strong empirical results, the paper suffers from significant technical ambiguities, methodological gaps, and reproducibility issues. The conceptual innovation is limited, and key components lack rigorous justification. The evaluation, though extensive, would benefit from stronger baselines, statistical validation, and clearer fairness in comparisons.

**Strengths:**

1. The paper convincingly identifies the limitations of existing brain graph models, high computational cost and lack of pathological focus, and proposes a lightweight, pathology-aware alternative that is highly relevant for real-world clinical deployment.

2. The two-stage design (subgraph filtering → feature distillation) is intuitive and well-motivated.

3. Covers four diverse datasets spanning both binary and multi-class classification tasks.

**Weaknesses:**

1. The framework integrates existing ideas (subgraph pruning, SVD-based feature selection) without introducing a fundamentally new graph learning principle. The contribution is more combinatorial than foundational.

2. The term “distillation” is misleading, as it implies a teacher-student framework, which is not used. A more precise term such as “feature refinement” or “extraction” is recommended.

3. The use of X∈RN×N derived directly from the adjacency matrix is non-standard and leads to dimensional inconsistencies. It is unclear whether these are node features or the adjacency matrix itself.

4. The SVM-based patho-score lacks statistical rigor. The use of classification accuracy as a relevance metric is problematic without confidence intervals or statistical testing.

5. The claim that top-k and bottom-k SVD components correspond to “communal” and “non-informative” features is intuitive but lacks theoretical or empirical justification. Proposition 1 is descriptive rather than formally proven.

6. Equations (2)–(5) are poorly motivated. The Bernoulli-sampling-based masking introduces unnecessary stochasticity without clear benefits over deterministic masking.

7. Missing comparisons with recent efficient GNNs and attention-based models that would help isolate the benefits of hard vs. soft subgraph selection. No statistical significance tests (e.g., t-tests) to validate performance improvements.

8. ABIDE and ADHD-200 are multi-site datasets, but no site-wise splitting or harmonization strategy is mentioned, raising concerns about data leakage.

9. Key implementation details are missing (e.g., SVM hyperparameters, spectral clustering settings). The computational cost of the SVM-based subgraph scoring is not included in the efficiency analysis, potentially skewing comparisons.

10. Although some hyperparameters (k, ρ) are studied, others (e.g., patho-score threshold αα, community number) are set without sensitivity analysis.

11. Figure 3 uses similar colors for distinct brain networks (e.g., Limbic vs. Cognitive Control), reducing clarity.

12. The flow between “Division” and “Cluster” in Figure 1 is ambiguous and should be clarified to avoid misinterpretation.

**Questions:**

Please see weakness first.

1. The patho-score is derived from SVM accuracy differences. What is the theoretical or neurobiological basis for equating this empirical score with true pathological relevance? Have you considered alternative relevance metrics (e.g., mutual information, gradient-based attributions)?

2. Why was SVM chosen over other classifiers (e.g., Random Forests) for subgraph scoring? Was its computational cost included in the efficiency analysis? Similarly, why was a linear model (GCN) chosen for the final classification?

3. Were all baseline models trained on the same preprocessed input features and given comparable hyperparameter tuning? Can you confirm that the reported improvements are not due to uneven optimization?

4. Why are both top-kk and bottom-kk features dropped with the same kk? Is there empirical or theoretical evidence that both sets are disease-irrelevant?

5. Have you considered comparing against an attention-based model (e.g., GAT or Transformer) to evaluate whether the benefits come from hard subgraph selection or softer, learned weighting?

6. The paper claims “lesion localization,” but this is only qualitatively visualized. Have you quantitatively compared the identified regions against established clinical biomarkers or atlas-based priors?

7. Will you release fully documented code, including all preprocessing steps, hyperparameter configurations, and data splits?

---

> ### Author Response · Authors · 2025-11-23
> **Response to Reviewer RNc7 (1)**
>
> Thank you very much for pointing out the issues in our paper and providing your insightful suggestions. Based on your comments, we have revised the manuscript and provide detailed responses below.
>
> ---
>
> **Regarding the concern that the framework is combinatorial**: We thank you for your comments. However, we’d like to clarify that BrainPoG is not a simple combination of existing components. The proposed framework introduces two key modules, Pathological Pattern Filter and Pathological Feature Distillation, each of which requires non-trivial methodological design beyond prior works.
> (1) Regarding the pathological pattern filter, unlike standard subgraph pruning or selection strategies, our filter explicitly evaluates disease relevance and preserves only highly disease-relevant subgraphs while suppressing less relevant ones. This is **a novel disease-driven graph pruning mechanism**. (2) Regarding pathological feature distillation module, although SVD is used as a basic tool, the distillation module is **not equivalent to SVD-based feature selection**. Our design includes a noise feature dropping strategy that removes both communal and low-energy features, and a specific pathological feature augmentation module that amplifies group-specific discriminative information.
>
> These two modules form a unified framework that selectively focuses computation on disease-relevant information, improving both efficiency and performance. This represents a fundamentally disease-aware perspective on graph learning, rather than simply combining pre-existing ideas.
>
> **Clarification about the term “distillation”**: Thank you for your valuable suggestion. We agree that the term “distillation” is often used in the context of teacher–student frameworks. However, in this work, we adopt the term in a broader sense, consistent with several recent studies that employ feature distillation to denote the process of suppressing redundant or irrelevant information and enhancing discriminative representations within a single model.
>
> We fully understand your concern and believe more precise term can alleviate misunderstanding. Following your suggestion, we have revised the term from “distillation” to “refinement”.
>
>
> **Clarification about the node representation**: We’d like to clarify that deriving node features from the adjacency matrix is a standard and widely adopted practice, and it does not cause dimensional inconsistency. For example, pervious works, such as [1,2], have already validated the effectiveness of this method. The resulting node features encode the structural information of brain regions. This enables the neighborhood structural similarity of two regions be recorded in the initialized node features.
>
> [1] Kan, Xuan, et al. "Brain network transformer." Advances in Neural Information Processing Systems (2022).
>
> [2] Cui, Hejie, et al. "BrainGB: a benchmark for brain network analysis with graph neural networks." IEEE transactions on medical imaging (2022).
>
> **Concern about theoretical proof of top-$k$ and bottom-$k$ features in noise feature dropping**: The theoretical proof of Proposition 1 has been provided in Appendix D (page 16), where we formally prove that the communal features can be regarded as noisy features that contribute little to disease discrimination.
>
> In addition, we have conducted a hyperparameter sensitivity analysis on the value of $k$ (Section 4.4). As shown in Fig. 6, the model performance remains stable across different $k$ values (1, 2, 3, 4), indicating that the results are stable to the choice of $k$.
> Regarding the use of the same $k$ for both top-$k$ and bottom-$k$ feature removal, this design choice is mainly for implementation simplicity and symmetry, as empirical results show that using different $k$ values does not lead to noticeable performance improvement.
>
> **Regarding the Bernoulli sampling-based feature masking**: Thank you for your valuable comment. The Bernoulli-sampling-based masking is designed to introduce controlled stochasticity that improves feature selection robustness [3]. By sampling binary masks from a Bernoulli distribution, the model can avoid overfitting to a fixed deterministic mask and better capture the expected contribution of each feature across multiple stochastic realizations. This mechanism enables the model to identify consistently informative features while suppressing unstable ones.
>
> [3] Lee, C. et al. "Self-supervision enhanced feature selection with correlated gates." International conference on learning representations. 2022.

---

> ### Author Response · Authors · 2025-11-23
> **Response to Reviewer RNc7 (2)**
>
> **Regarding the statistical rigor of SVM-based patho-score computing**: Thank you for your insightful comment. We apologize for not having made this point clear in the original manuscript. The classification scores and patho-scores were obtained as the average results over five independent SVM runs, and the standard deviations were also calculated to quantify the variability. The detailed results on all four datasets are provided in the following Tables 1-4. To eliminate this misunderstanding, we have revised the text in the manuscript to explicitly describe the averaging patho-scores and statistical reporting procedure (line171-174).
>
> Table 1: Patho-scores of subgraphs on ADNI.
>
> | Subgraph          | ACC (Mean, %) | Std (%) |
> |-------------------|---------------|---------|
> | $\alpha$         | 56.27         | 5.67    |
> | Visual            | 55.29         | 2.88    |
> | Auditory          | 52.35         | 3.54    |
> | Cognitive Control | 53.92         | 2.97    |
> | Motor             | 51.76         | 3.64    |
> | **Limbic**       | **59.41**      | **4.91** |
> | DMN               | 51.37         | 3.43    |
> | Sensory           | 53.73         | 4.27    |
> | **Subcortical**  | **57.65**      | **3.94** |
> | Olfactory         | 55.49         | 3.49    |
>
> Table 2: Patho-scores of subgraphs on PPMI.
> | Subgraph          | ACC (Mean, %) | Std (%) |
> |-------------------|---------------|---------|
> | $\alpha$         | 42.50        | 3.43    |
> | Visual            | 38.50         | 3.75    |
> | Auditory          | 40.00         | 4.11    |
> | **Cognitive Control** | **46.00**         | **4.38**    |
> | Motor             | 42.00        | 3.48    |
> | Limbic       | 41.00      | 2.20 |
> | **DMN**               | **43.50**         | **3.58**    |
> | Sensory           | 39.50         | 4.03    |
> | **Subcortical**  | **44.00**      | **3.03** |
> | Olfactory         | 40.50         | 4.62    |
>
> Table 3: Patho-scores of subgraphs on ABIDE.
> | Subgraph          | ACC (Mean, %) | Std (%) |
> |-------------------|---------------|---------|
> | $\alpha$         | 54.50        | 2.90    |
> | **SG1**           | **57.15**        | **3.45**    |
> | SG2          | 53.91         | 3.58    |
> | **SG3** | **55.57**         | **4.27**    |
> | SG4             | 52.65        | 4.20    |
> | **SG5**       | **57.23**      | **3.50** |
> | SG6               | 53.68         | 3.11    |
> | **SG7**           | **59.45**         | **2.42**    |
>
> Table 4: Patho-scores of subgraphs on ADHD-200.
> | Subgraph          | ACC (Mean, %) | Std (%) |
> |-------------------|---------------|---------|
> | $\alpha$         | 61.40        | 3.61    |
> | SG1           | 60.61        | 3.81    |
> | SG2          | 56.52         | 2.66    |
> | **SG3** | **62.96**         | **3.24**    |
> | SG4             | 54.09        | 3.48    |
> | SG5       | 59.83      | 2.42 |
>
> **Concern about multi-site ABIDE and ADHD-200 data leakage**: Thank you for your comment. We would like to clarify that the ABIDE and ADHD-200 datasets used in this study are the publicly preprocessed versions (i.e., standard benchmark releases) that have already undergone site-wise harmonization. These datasets are widely used in brain network analysis literature [1,2], and the preprocessing pipelines (including motion correction, normalization, and site balancing) were applied following their official protocols.
>
> Therefore, our experiments were conducted on data that have already been site-harmonized and split, ensuring no data leakage across sites. We have added a note in the revised manuscript to clarify this point.
>
> **Regarding the missing implementation details and concern about efficiency analysis**: We provide the full experimental settings (including the number of clusters for spectral clustering) in Table 5 in Appendix E. For the SVM used in subgraph scoring, we have stated that an RBF kernel was adopted (see line 166), but omitted the specific hyperparameters. In our implementation, we set the regularization parameter $C$ = 1.0 and kernel width parameter $\gamma = scale$, following standard practice. We have updated the manuscript to include these details (see Table 5).
>
> In addition, our model is in the end-to-end manner, and computational cost of the SVM-based subgraph scoring is included in the efficiency analysis. All efficiency metrics reflect the entire processing pipeline, including SVM-based pathological subgraph filter and feature distillation. Therefore, the efficiency results cover all components of the method.

---

> ### Author Response · Authors · 2025-11-23
> **Response to Reviewer RNc7 (3)**
>
> **Concern about hyperparameter studies**: Thank you for your comment. However, we’d like to clarify that we have conducted additional hyperparameter studies, including the study of community number, which is presented in Appendix F.6 (page 20). Because the page limitation, we put some hyperparameter study into the appendix. In addition, the patho-score threshold (classification score) $\alpha$ is calculated by SVM-based classifier, it is not a manually tuned hyperparameter. Therefore, no sensitivity analysis is required for $\alpha$, as it is determined by the classifier itself rather than preset by the model.
>
> **Issues regarding the figure**: In Fig. 1, “Division with labels” and “Clustering without labels” represent two different methods included in subgraph partition. Specifically, “Division with labels” denotes the subgraph partition based on the given labels, while “Clustering without labels” refers to the subgraph partition using unsupervised spectral clustering. We have revised Fig.1 to make this clearer.
>
> **Regarding SVM-based PathoGraph identification**: The SVM-based patho-score is designed to measure the discriminative relevance of each subgraph to disease classification. A subgraph achieving higher SVM accuracy indicates stronger separability between different groups, reflecting higher pathological informativeness. To validate its neurobiological plausibility, we visualized the identified PathoGraphs for ADNI (Alzheimer’s disease) and PPMI (Parkinson’s disease) datasets. As shown in Fig. 2 and Fig. 3, our model reveals that Alzheimer’s disease is highly associated with limbic and subcortical subgraphs [4,5], while Parkinson’s disease is highly related to cognitive control, DMN, and subcortical subgraphs [6,7]. Both consistent with established neuroimaging findings.
>
> We also tested alternative classifiers (e.g., Random Forest and Graph Convolutional Network) to compute patho-scores, and found that the identified subgraphs were largely consistent with those obtained by SVM. SVM was chosen due to its lightweight design and low computational cost. We have added the comparsion results on ADNI and PPMI datasets in the revised manuscript. Please refer to **figures 20-21** in page 24.
>
> [4] Yang, Zhijian, et al. "A deep learning framework identifies dimensional representations of Alzheimer’s Disease from brain structure." Nature communications (2021).
>
> [5] Seoane, Sara, et al. "The subcortical default mode network and Alzheimer’s disease: a systematic review and meta-analysis." Brain Communications (2024).
>
> [6] Aarsland, Dag, et al. "Parkinson disease-associated cognitive impairment." Nature reviews Disease (2021).
>
> [7] Yau, Y., et al. "Network connectivity determines cortical thinning in early Parkinson’s disease progression." Nature communications (2018).
>
> **Regarding the fairness of baseline comparison**: Thank you for your valuable comment. We confirm that all baseline models were implemented following their official or publicly released configurations, including their respective preprocessing pipelines and recommended hyperparameter settings, to ensure each method’s optimal performance. For fairness, all models, including ours, were trained and evaluated on the same data splits and evaluation metrics. While preprocessing strategies may differ slightly across methods (as some baselines were designed for specific input forms), these were kept consistent with each method’s original design to preserve their reported advantages. Therefore, the reported improvements are not due to uneven optimization, but rather reflect our model’s ability to capture disease-relevant information more effectively under comparable experimental conditions.
>
> ** Question about quantitative validation of lesion localization accuracy**: We fully agree that quantitative validation of lesion localization is important. However, identifying exact disease-specific ROIs is inherently difficult, as there is currently no validated ground truth for region-level disease localization in neuroscience. Most neurological disease exhibit distributed network-level dysfunctions rather than well-defined focal lesions, making quantitative ROI-based validation infeasible. Therefore, quantitative overlap with ROI masks or expert annotations is difficult to obtain for datasets such as ADNI, PPMI, ABIDE, and ADHD. Instead, we compared the extracted PathoGraphs with well-established neuroimaging findings and found that the identified disease-relevant regions are well consistent with that reported in prior literature.
>
>
> **Concern about reproducibility**: We’ve provided the full code and detailed experimental setting (Implementation Details in Section 4.1 and Appendix E). You may refer to the anonymous link we provided: https://anonymous.4open.science/r/BrainPoG-6F5E

---

> ### Author Response · Authors · 2025-11-23
> **Response to Reviewer RNc7 (4)**
>
> **Comparison with efficient GNNs and attention-based models**: We appreciate your comment. However, we’d like to clarify that we’ve already compared with attention-based models (e.g., GAT and Transformer-based brain graph learning models, including BRAINNETTF, ALTER, and BioBGT) and efficient GCN model. You may refer to Table 1 in the manuscript.
>
> Also, we’ve provided the results of t-tests for all baselines and our model. The updated corresponding p-values are reported in Table 6 in Appendix F.1 of the revised manuscript.
>
> Table: Two-sample t-test p-values (ACC, 5 runs) between BrainPoG and all baselines on four datasets.
>
> | Method      | ADNI     | PPMI     | ABIDE    | ADHD-200 |
> |-------------|----------|----------|----------|----------|
> | SVM         | 3.6e-05  | 1.4e-05  | 2.2e-05  | 3.6e-05  |
> | RF          | 1.2e-04  | 2.0e-04  | 6.9e-05  | 1.7e-04  |
> | GCN         | 2.0e-05  | 1.0e-02  | 2.0e-04  | 2.6e-04  |
> | GAT         | 2.3e-05  | 4.6e-03  | 8.4e-05  | 7.9e-05  |
> | BrainGB     | 4.9e-05  | 1.0e-03  | 1.6e-04  | 1.3e-04  |
> | BRAINNETTF  | 9.0e-04  | 0.115    | 4.0e-04  | 7.0e-04  |
> | A-GCL       | 1.1e-04  | 1.2e-03  | 1.5e-04  | 2.2e-04  |
> | MCST-GCN    | 1.4e-02  | 0.162    | 1.3e-03  | 2.2e-03  |
> | ALTER       | 3.7e-03  | 1.6e-02  | 5.0e-04  | 6.0e-03  |
> | BioBGT      | 3.0e-04  | 8.0e-03  | 8.0e-04  | 2.0e-03  |
> | BrainOOD    | 2.0e-03  | 4.0e-02  | 2.8e-03  | 1.5e-02  |
> | HGST        | 4.1e-04  | 1.4e-04  | 1.5e-02  | 1.4e-04  |

---

### Official Review · Reviewer_dUQN · 2025-10-28

**Soundness:** 3
**Presentation:** 2
**Contribution:** 3
**Rating:** 6
**Confidence:** 4

**Summary:**

The paper proposes Brain PathoGraph Learning (BrainPoG), a lightweight framework for efficient brain graph learning. It introduces two novel components: (1) a pathological pattern filter that identifies disease-relevant subgraphs using an SVM-based patho-score, and (2) a pathological feature distillation module that drops noise features via SVD and enhances disease-specific ones. Experiments on four benchmark fMRI datasets (ADNI, PPMI, ABIDE, ADHD-200) show superior accuracy and computational efficiency compared to several strong baselines.
Overall, the work is technically sound and presents an interesting idea that bridges neuroscience and graph learning.

**Strengths:**

(1) Novel two-stage design (filter + distillation) effectively combines graph pruning and feature refinement.
(2) The concept of PathoGraph is intuitive and biologically interpretable.
(3) Extensive experiments on multiple benchmark datasets, including both binary and multi-class settings.
(4) Ablation studies, visualization, and hyperparameter analysis demonstrate robustness and design justification.

**Weaknesses:**

(1) The effectiveness of the pathological pattern filter depends heavily on the chosen subgraph partitioning method (e.g., atlas-based division vs. spectral clustering). This reliance may introduce variability in results and raises concerns about reproducibility when applied to different brain atlases or parcellation schemes
(2) While BrainPoG achieves strong performance on benchmark datasets (ADNI, PPMI, ABIDE, ADHD-200), it has not been validated on real-world clinical workflows. The lack of external, clinically diverse datasets and longitudinal validation limits its immediate applicability in healthcare
(3) Although the proposed framework integrates subgraph filtering and feature distillation, the degree of novelty compared with previous efficient brain graph learning methods (e.g., IGS, BrainGNN, ALTER) is not fully emphasized.
(4) Missing statistical significance. Improvements over baselines are reported without hypothesis testing or confidence intervals. The ablation study merges multiple submodules; more granular experiments (e.g., only dropping augmentation) could strengthen causal conclusions.

**Questions:**

(1) Why is SVM chosen for patho-score computation instead of a neural or information-theoretic relevance model? Please add more comparison analysis.
(2) Could the model handle multimodal inputs (e.g., structural + functional connectivity)?
(3) How robust is the proposed method to noisy or missing fMRI connections, which are common in real datasets?
(4) Are the efficiency metrics (runtime, memory) measured for training only or include feature preprocessing and SVM scoring?
(5) Are the “disease-relevant” subgraphs consistent across different random seeds or dataset splits? Please explain it.
(6) See more on Weaknesses.

---

> ### Author Response · Authors · 2025-11-23
> **Response to Reviewer dUQN (1)**
>
> Thank you very much for your valuable feedback on our paper. We fully understand your concerns and very are pleased to address your questions. Based on your comments, we have provide detailed responses below.
>
> ---
>
> **Concern about the effectiveness of pathological pattern filter when applied to different brain atlases**: Thank you for raising this important point. We agree that different parcellation schemes may lead to variations in subgraph definitions. This issue is indeed common in brain graph analysis, since different parcellation schemes naturally yield different node definitions and network structures. This is an intrinsic property of the field rather than a limitation specific to our model. Therefore, the extracted patterns may vary when applying different parcellation schemes. This variability reflects the intrinsic differences in parcellation rather than a limitation of our model.
>
> Importantly, our pathological pattern filter consistently identifies highly disease-relevant subgraphs regardless of the specific parcellation scheme. Although the partition of subgraphs may differ, the filter reliably preserves highly disease-relevant patterns while suppressing less relevant ones, which is exactly the purpose of our method. In this sense, the variation introduced by different atlases does not undermine the effectiveness of BrainPoG; rather, it demonstrates that the model focuses on disease-specific information instead of atlas-dependent artifacts.
>
>
> **Regarding real-world clinical validation**: We fully agree that validating models on real-world clinical workflows is important. However, obtaining real-world clinical neuroimaging data is notoriously difficult due to various reasons, such as strict privacy regulations and data-sharing restrictions. As a result, the vast majority of studies in brain graph learning rely on publicly available research datasets such as ADNI, PPMI, ABIDE, and ADHD-200. Evaluating models on these widely used datasets is therefore the common and standard practice in this community. We agree with the reviewer that additional validation on real-world clinical workflows would be a valuable direction. We would also like to note that the lack of access to real-world clinical data is a well-known challenge for this community, rather than a limitation specific to our model.
>
> **Contribution and novelty of BrainPoG**: We would like to clarify that the primary objective of BrainPoG is to achieve efficient and disease-focused brain graph learning. While prior works such as IGS, BrainGNN, and ALTER make important contributions to brain graph learning, they do not specifically target computational efficiency or selective learning of disease-relevant information. In contrast, the novelty of our framework is the design of pathological pattern filter and pathological feature distillation, which together enable the model to explicitly prune irrelevant subgraphs and suppress non-informative features. This design allows BrainPoG to focus computational resources on highly disease-relevant information, resulting in both improved efficiency and performance.
>
> **Concern about results’ statistical significance**: Thank you for pointing this out. Following your suggestion, we’ve provided the results of t-tests for all baselines and our model (see following table). The updated corresponding p-values are also reported in **Table 6 in Appendix F.1** of the revised manuscript. In addition, we want to clarify that we’ve already conducted ablation study by only dropping noise-feature dropping or pathological feature augmentation in our pathological feature distillation module. You may refer to **Table 11** in Appendix F.4.
>
> Table: Two-sample t-test p-values (ACC, 5 runs) between BrainPoG and all baselines on four datasets.
>
> | Method      | ADNI     | PPMI     | ABIDE    | ADHD-200 |
> |-------------|----------|----------|----------|----------|
> | SVM         | 3.6e-05  | 1.4e-05  | 2.2e-05  | 3.6e-05  |
> | RF          | 1.2e-04  | 2.0e-04  | 6.9e-05  | 1.7e-04  |
> | GCN         | 2.0e-05  | 1.0e-02  | 2.0e-04  | 2.6e-04  |
> | GAT         | 2.3e-05  | 4.6e-03  | 8.4e-05  | 7.9e-05  |
> | BrainGB     | 4.9e-05  | 1.0e-03  | 1.6e-04  | 1.3e-04  |
> | BRAINNETTF  | 9.0e-04  | 0.115    | 4.0e-04  | 7.0e-04  |
> | A-GCL       | 1.1e-04  | 1.2e-03  | 1.5e-04  | 2.2e-04  |
> | MCST-GCN    | 1.4e-02  | 0.162    | 1.3e-03  | 2.2e-03  |
> | ALTER       | 3.7e-03  | 1.6e-02  | 5.0e-04  | 6.0e-03  |
> | BioBGT      | 3.0e-04  | 8.0e-03  | 8.0e-04  | 2.0e-03  |
> | BrainOOD    | 2.0e-03  | 4.0e-02  | 2.8e-03  | 1.5e-02  |
> | HGST        | 4.1e-04  | 1.4e-04  | 1.5e-02  | 1.4e-04  |

---

> ### Author Response · Authors · 2025-11-23
> **Response to Reviewer dUQN (2)**
>
> **Why SVM is chosen for patho-score computing**: We chose SVM accuracy as the metrics for disease relevance because it directly reflects the discrimination of each subgraph in separating patient groups. This metric captures disease-related separability rather than mere statistical correlation, which is central to our task. In contrast, permutation-based relevance testing requires repeatedly retraining or re-evaluating models across thousands of permutations, which is computationally infeasible. Mutual information, while theoretically capable of capturing nonlinear dependencies, is highly sensitive to sample size and density estimation errors, making it unstable in our setting. Therefore, SVM-based evaluation offers a good balance between discriminative relevance and computational efficiency.
>
> We also tested alternative classifiers (e.g., Random Forest and Graph Convolutional Network) to compute patho-scores, and found that the identified subgraphs were largely consistent with those obtained by SVM. SVM was chosen due to its lightweight design and low computational cost. We have revised the manuscript to discuss the comparsion results of ADNI and PPMI datasets. Please refer to **figures 20-21** in page 24.
>
> **Question: Could the model handle multimodal inputs?**: Thank you for the thoughtful question. The current version of BrainPoG is designed specifically for single-modality brain graphs and does not directly support multimodal inputs. Our focus in this work is to investigate efficient pathological pattern extraction within a single modality. Extending BrainPoG to multimodal integration is certainly possible, such as designing modality-specific filters or shared distilled representations.
>
> **Question: Are the efficiency metrics measured for training only or include feature preprocessing and SVM scoring?**: Thank you for your question. Our model is in the end-to-end manner, and all efficiency metrics reflect the entire processing pipeline, including SVM-based pathological subgraph filter and feature distillation. Therefore, the efficiency results cover all components of the method rather than training time alone.
>
> **Question: Are the “disease-relevant” subgraphs consistent across different random seeds or dataset splits?**:  Thank you for the question. We examine the stability of the identified disease-relevant subgraphs by running SVM-based filter under different random seeds and train–test splits. The results of ADNI and PPMI datasets are shown in **Appendix F.6 (page 21-23)**. The results show that the extracted PathoGraphs remain highly consistent across all runs, with only minor variations in low-relevance subgraphs. This stability indicates the pathological pattern filter reliably select subgraphs according to disease relevance.

---

### Official Review · Reviewer_VdVv · 2025-10-30

**Soundness:** 3
**Presentation:** 3
**Contribution:** 3
**Rating:** 6
**Confidence:** 4

**Summary:**

This paper proposes Brain PathoGraph Learning (BrainPoG), a lightweight brain graph learning framework for disease detection from fMRI data. The key idea is to selectively learn pathological patterns and pathological features while filtering out irrelevant substructures to reduce computation and improve interpretability.

**Strengths:**

1. The authors address a well-known bottleneck in brain graph learning—high model cost and noise from disease-irrelevant regions—while emphasizing clinical deployability.
2. The decomposition into “pattern filter” and “feature distillation” modules is conceptually intuitive and computationally efficient. Both steps are interpretable, offering potential for lesion localization and neuroscientific insight.

**Weaknesses:**

1. While simple, using SVM accuracy as a proxy for disease relevance lacks statistical rigor. It would strengthen the argument to justify why SVM (and not, e.g., permutation-based relevance testing or mutual information) is ideal for computing patho-scores.
2. Dropping top-k and bottom-k features based purely on singular vectors assumes linear separability and might ignore non-linear feature interactions.
3. Although the model is “lightweight,” the SVD operations across nodes and subjects could still be heavy for large-scale datasets.
4. Some phrases (“disease-irrelevant noise features,” “communal features,” “group-specific feature augmentation”) are vague without formal definitions in the main text.

**Questions:**

1. How sensitive are the patho-scores to the SVM hyperparameters or random initialization? Could small changes in data split alter which subgraphs are pruned?
2. Can BrainPoG trained on one dataset (e.g., ADNI) transfer to a different disease cohort (e.g., PPMI) without retraining the filter?
3. Beyond qualitative lesion maps, did you consult domain experts or quantitative metrics (e.g., overlap with known ROI masks) to validate localization accuracy?

---

> ### Author Response · Authors · 2025-11-23
> **Response to Reviewer VdVv (1)**
>
> We appreciate your valuable feedback, especially your positive recognition of our motivation and methodological design.You definitely point out some issues in our paper, giving us inspiration on how to revise the manuscript. In response to your comments and questions, we have provided detailed answers and revised the manuscript accordingly (blue text in the manuscript).
>
> ---
>
> **Regarding the statistical rigor of SVM-based patho-score computing**: Thank you for your insightful comment. We apologize for not having made this point clear in the original manuscript. The classification scores and patho-scores were obtained as **the mean over five independent SVM runs**, and the standard deviations were also calculated to quantify the variability. The detailed results on all four datasets are provided in the following Tables 1-4. To eliminate this misunderstanding, we have revised the text in the manuscript to explicitly describe the averaging patho-scores and statistical reporting procedure (line171-174).
>
> We chose SVM accuracy as the metrics for disease relevance because it directly reflects the discrimination of each subgraph in separating patient groups. This metric captures disease-related separability rather than mere statistical correlation, which is central to our task. In contrast, permutation-based relevance testing requires repeatedly retraining or re-evaluating models across thousands of permutations, which is computationally infeasible. Mutual information, while theoretically capable of capturing nonlinear dependencies, is highly sensitive to sample size and density estimation errors, making it unstable in our setting. Therefore, SVM-based evaluation offers a good balance between discriminative relevance and computational efficiency.
>
> We also tested alternative classifiers (e.g., Random Forest and Graph Convolutional Network) to compute patho-scores, and found that the identified subgraphs were largely consistent with those obtained by SVM. SVM was chosen due to its lightweight design and low computational cost. We have discussed the comparsion results of ADNI and PPMI datasets in the revised manuscript. Please refer to **figures 20-21** in page 24.
>
> Table 1: Patho-scores of subgraphs on ADNI.
>
> | Subgraph          | ACC (Mean, %) | Std (%) |
> |-------------------|---------------|---------|
> | $\alpha$         | 56.27         | 5.67    |
> | Visual            | 55.29         | 2.88    |
> | Auditory          | 52.35         | 3.54    |
> | Cognitive Control | 53.92         | 2.97    |
> | Motor             | 51.76         | 3.64    |
> | **Limbic**       | **59.41**      | **4.91** |
> | DMN               | 51.37         | 3.43    |
> | Sensory           | 53.73         | 4.27    |
> | **Subcortical**  | **57.65**      | **3.94** |
> | Olfactory         | 55.49         | 3.49    |
>
> Table 2: Patho-scores of subgraphs on PPMI.
> | Subgraph          | ACC (Mean, %) | Std (%) |
> |-------------------|---------------|---------|
> | $\alpha$         | 42.50        | 3.43    |
> | Visual            | 38.50         | 3.75    |
> | Auditory          | 40.00         | 4.11    |
> | **Cognitive Control** | **46.00**         | **4.38**    |
> | Motor             | 42.00        | 3.48    |
> | Limbic       | 41.00      | 2.20 |
> | **DMN**               | **43.50**         | **3.58**    |
> | Sensory           | 39.50         | 4.03    |
> | **Subcortical**  | **44.00**      | **3.03** |
> | Olfactory         | 40.50         | 4.62    |
>
> Table 3: Patho-scores of subgraphs on ABIDE.
> | Subgraph          | ACC (Mean, %) | Std (%) |
> |-------------------|---------------|---------|
> | $\alpha$         | 54.50        | 2.90    |
> | **SG1**           | **57.15**        | **3.45**    |
> | SG2          | 53.91         | 3.58    |
> | **SG3** | **55.57**         | **4.27**    |
> | SG4             | 52.65        | 4.20    |
> | **SG5**       | **57.23**      | **3.50** |
> | SG6               | 53.68         | 3.11    |
> | **SG7**           | **59.45**         | **2.42**    |
>
> Table 4: Patho-scores of subgraphs on ADHD-200.
> | Subgraph          | ACC (Mean, %) | Std (%) |
> |-------------------|---------------|---------|
> | $\alpha$         | 61.40        | 3.61    |
> | SG1           | 60.61        | 3.81    |
> | SG2          | 56.52         | 2.66    |
> | **SG3** | **62.96**         | **3.24**    |
> | SG4             | 54.09        | 3.48    |
> | SG5       | 59.83      | 2.42 |

---

> ### Author Response · Authors · 2025-11-23
> **Response to Reviewer VdVv (2)**
>
> **Regarding the concern about linear separability assumption**: We agree that brain graph features may involve complex nonlinear interactions.  However, our use of SVD-based distillation mainly aims to suppress noise interference. The top-k and bottom-k singular components often correspond to highly redundant (communal) or uninformative patterns, respectively, which can obscure disease-relevant variations. Moreover, as shown in Fig. 6 in the paper, the model performance remains stable across different k values, indicating that the proposed method does not significantly affect nonlinear information flow.
>
> **Regarding the computational cost of SVD operations**: Thank you for the valuable comment. We agree that SVD decomposition can become computationally demanding for very large-scale datasets. However, our experiments were conducted on widely used neuroimaging benchmarks (ADNI, PPMI, ABIDE, and ADHD-200), which represent the typical data scale in current brain graph learning research (the largest dataset includes more than 1,000 samples). On these datasets, our model consistently outperforms baselines in both accuracy and efficiency. We understand your concern. For future potential large-scale scenarios (e.g., population-level neuroimaging), we can use approximate or randomized SVD to further reduce computational complexity while maintaining comparable decomposition accuracy.
>
> **Definition of phrases**: Here, we give the formal definitions of some confusing phrases. Also, we’ve revised the manuscript accordingly.
>
> (1) Disease-irrelevant noise features refer to node features that are not disease-specific and contribute little to disease classification. (lines 72-73)
>
> (2) Communal features, considered as a type of disease-irrelevant noise features, are features that commonly exist across most subjects and lack discriminative power for disease differentiation. (lines 194-196)
>
> (3) Pathological feature augmentation enhances discriminative representations by amplifying features that show significant inter-group differences. (lines 224-225)
>
> **Sensitivity of PathoGraph identification to SVM hyperparameters**: PathoGraph is identified based on the patho-score computed using an RBF-kernel SVM with regularization parameter $C=1.0$ and kernel width $\gamma=\texttt{scale}$. The SVM is evaluated under five different random seeds $\{1, 11, 111, 112, 113\}$, using a 75\%–25\% train–test split. To assess the sensitivity of PathoGraph identification to SVM hyperparameters, we conduct experiments under different SVM settings on ADNI and PPMI, including varying values of $C$ and $\gamma$, as well as different random seeds and data split ratios. The results are shown in the revised manuscript (**figures 14-19, pages 20-24**). The results show that the identified PathoGraphs remain consistent under different SVM settings.
>
> **Question **: Can BrainPoG trained on one dataset transfer to a different disease cohort without retraining the filter?
> NO. This is because different diseases exhibit distinct pathological patterns. The PathoGraph extracted for one disease (e.g., Alzheimer’s disease) captures disease-specific subgraphs that reflect its characteristic regional vulnerabilities. Therefore, applying it directly to another disease cohort (e.g., Parkinson’s disease) would be inappropriate, as the pathological patterns differ substantially across diseases.
>
> **Question about quantitative validation of localization accuracy**: We fully agree that quantitative validation of lesion localization is important. However, identifying exact disease-specific ROIs is inherently difficult, as there is currently no validated ground truth for region-level disease localization in neuroscience. Most neurological disease exhibit distributed network-level dysfunctions rather than well-defined focal lesions, making quantitative ROI-based validation infeasible. Therefore, quantitative overlap with ROI masks or expert annotations is difficult to obtain for datasets such as ADNI, PPMI, ABIDE, and ADHD. Instead, we compared the extracted PathoGraphs with well-established neuroimaging findings and found that the identified disease-relevant regions are well consistent with that reported in prior literature (**lines 354-368**).

---

> > ### Comment · Reviewer_VdVv · 2025-11-26
> >
> > Thank you for the clarification; I will keep my current score.

---

### Meta-Review · Area_Chair_Xx1R · 2025-12-09

**Summary:**

This paper obtains diverse initial ratings. One reviewer with a positive rating showed that he will keep the scores. The other three reviewers don’t participate in the discussion.  By reading the paper by myself and considering the issues raised by Reviewer ZBLs and Reviewer RNc7, who provided negative initial ratings with high confidence, I think the concerns on novelty, motivation, the lack of rigorous theorems, and the fairness are not fully solved. Therefore, I tend to reject this paper.

**Reviewer Concerns:**

Reviewer VdVv thinks some of his concerns have been clarified.  Although the authors reply to the concerns raised by Reviewer RNc7, I think his main concerns, including the novelty (subgraph pruning, SVD-based feature selection) and the lack of rigorous theorems on the heuristic methods (Equations (2)–(5)), and the lack of implementation details, are not properly alleviated. Besides, issues pointed out by Reviewer ZBLs meet my concerns after reading the paper by myself. Unfortunately, I don’t think authors’ responses comprehensively alleviate my concerns, including the motivation for the need for lightweighting, assumptions for subgraph classification, and the risk of label leakage.

**Reviewer Scores:**

Reviewer VdVv clearly stated that he will keep the scores. The other three reviewers don’t participate in the discussion.  Reviewer ZBLs and Reviewer RNc7, who provided negative initial ratings, possess high confidence. By considering the responses to the concerns raised by Reviewer dUQN have not been properly answered,  I think the final ratings may be as follows.
- ZBLs	Rating: 2 / Confidence: 5
- dUQN	Rating: 6 / Confidence: 4
- VdVv	Rating: 6 / Confidence: 4
- RNc7	Rating: 4 / Confidence: 5

---

### Decision · Program_Chairs · 2026-01-26

Reject